# Entangled No More: Multi-Domain Decoupling for Robust Dynamic Graph Neural Networks

Youda Mo[1]  Chaobo He[1]  Junwei Cheng[2][1]  Peng Mei[1]  Quanlong Guan[3]

## Abstract

Dynamic graphs are pervasive in real-world systems, but their tightly entangled spatiotemporal evolution causes significant modeling challenges. Existing Dynamic Graph Neural Networks (DGNNs) lack a principled framework for systematically decoupling this multi-domain entanglement, raising two key problems: (i) representation drift caused by structural incompleteness, and (ii) signal distortion amplified by noise perturbation. These problems can accumulate over time, forming temporal redundancy that weakens robustness of DGNNs. In view of these, we propose DeR-Mamba (Decoupling for Robust Mamba), a multi-domain decoupling framework for robust DGNNs. To address (i), we develop the Multi-Particle Kernel Kalman observation field (MP–K²alman), which achieves spatial decoupling by sampling latent evolution paths in kernel subspaces and performing Kalman-style updates to estimate structural states. To address (ii), we design the Adversarial-aware Frequency Decoupling Module (AFDM), which performs frequency-domain decoupling and dynamic cross-frequency modulation to purify spectral signals. Finally, a self-consistent dynamic graph state-space system performs temporal decoupling to control redundancy, suppressing residual disturbances through discretized cross-time modeling and selective snapshot scanning. Extensive experiments on benchmark datasets with adversarial attacks validate its superior robustness.

## 1. Introduction

Dynamic graphs are ubiquitous in real-world systems, ranging from social and financial systems (He et al., 2023;

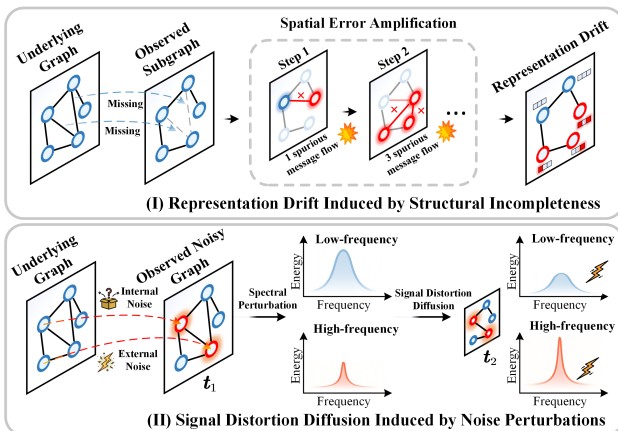

*Figure 1.* **Problem illustration.** Dynamic graphs often evolve under incomplete and noisy observations. **(I) Structural incompleteness** causes spurious messages and amplifies spatial errors, resulting in representation drift. **(II) Noise perturbations** induce signal distortion through spectral shifts and can diffuse over time.

Cheng et al., 2025; Liu et al., 2026) to traffic networks and knowledge graphs (Sun et al., 2025; Cao et al., 2025; Zhao et al., 2025). Dynamic Graph Neural Networks (DGNNs) have emerged as the predominant representation learning paradigm by modeling the evolving structures and time-varying node attributes (Li et al., 2024b; Chen et al., 2025; Xu et al., 2025). However, most DGNNs implicitly assume that the observed graph at each snapshot is complete and noise-free, which rarely holds in practice.

As illustrated in Figure 1, real-world dynamic graphs are imperfect observations and suffer from structural incompleteness and noise perturbations. Structural incompleteness, caused by sensor latency or sampling bias (Gu et al., 2025; Li et al., 2025a), yields missing edges that distort local connectivity. During message passing, these defects induce spurious flows, accumulate spatial errors, and manifest as representation drift (Xu et al., 2020; Li & Huang, 2023). Meanwhile, from a graph signal processing perspective, dynamic graphs are often corrupted by external environmental disturbances (Lin et al., 2022) and internal stochasticity (Wang et al., 2025a). Such noise amplifies non-stationary high-frequency components, perturbs spectral stationarity, and diffuses distortions across snapshots. As temporal evolution proceeds, these observational defects even will accumulate into temporal redundancy.

[1]School of Computer Science, South China Normal University
[2]China Electronic Product Reliability and Environmental Testing Research Institute [3]Department of Computer Science, Jinan University. Correspondence to: Chaobo He <hechaobo@foxmail.com>.

*Proceedings of the 43rd International Conference on Machine Learning*, Seoul, South Korea. PMLR 306, 2026. Copyright 2026 by the author(s).

Moreover, these imperfections never evolve in isolation, but are always tightly entangled. Specifically, structural deficiencies accumulated through temporal propagation intensify noise, drastically reshaping the feature spectral distribution and precipitating surges of high-frequency components. Such spectral deviations erode temporal dependencies and inevitably trigger cross-time-step resonance. Consequently, structural incompleteness, noise perturbations, and temporal redundancy are inextricably linked, forming a complex multi-domain entangled system (Kong et al., 2024). Existing DGNNs (Zhang et al., 2025; Fu & Ren, 2026) fail to account for this coupled nature, allowing errors from different domains to cascade and reinforce one another, which renders them unable to learn robust representations.

Aiming to these challenges above, we introduce DeR-Mamba, a robust dynamic graph representation framework that systematically decouples multi-domain entanglements. Specifically, to address (I), we integrate MP–K²alman, which models structural uncertainty via multi-particle sampling and kernel mappings, followed by a Kalman-style confidence-weighted update. To address (II), we incorporate AFDM, using wavelet-based decomposition and adaptive modulation to attenuate noise-dominant high-frequency responses while enhancing meaningful abrupt variations. To counteract temporal redundancy accumulated during dynamic evolution, we employ a selective state-space decoupling mechanism that filters redundant cross-time pathways and activates critical transitions, enabling stable modeling of global temporal dynamics.

In summary, our main contributions are as follows:

- We propose MP–K²alman, which utilizes multi-particle sampling and Kalman-style state updates to model structural uncertainty and achieve robust spatial decoupling, effectively mitigating representation drift caused by structural incompleteness.

- We develop AFDM, which leverages frequency-domain decoupling and dynamic cross-frequency modulation to adaptively suppress high-frequency spectral noise, enhance critical structural responses, and enable robust signal reconstruction.

- We design a selective state-space decoupling mechanism that models cross-step temporal dependencies and activates key states to capture global dynamics, effectively mitigating temporal redundancy and enabling robust temporal decoupling.

- DeR-Mamba is the first framework to systematically analyze spatial, spectral, and temporal perturbations in dynamic graph evolution. Extensive experiments across 11 adversarial scenarios validate its robustness.

## 2. Related Work

**Robust dynamic graph learning.** DGNNs model evolving node interactions but remain vulnerable to structural perturbations and noise-induced instability, leading to degraded robustness in real-world or adversarial environments (Lee et al., 2024). Recent advances in robust dynamic graph learning explore adversarial training and frequency- or wavelet-based representations (Li et al., 2025b; Zheng et al., 2025), but these methods typically address perturbations from a single perspective and rely on the assumption of clean and fully observed snapshots.

**Kalman filtering in graph-structured systems.** Kalman filtering provides minimum-variance estimation under perturbed observations, a principle relevant to graph-structured settings where latent states evolve under structural incompleteness and noise. Early graph-based variants, such as KalmanNet (Buchnik et al., 2024) and PKF (Liu et al., 2024; Wang et al., 2025a), show the promising performance, but are not applicable to dynamic graphs. In particular, the robustness challenge remains largely unexplored due to multi-domain entanglement induced by dynamic evolution.

**Graph modeling with state space models.** State Space Models (SSMs) enable efficient long-range dependency modeling, and selective variants such as S4 and Mamba (Gu & Dao, 2024) have motivated graph adaptations that convert non-Euclidean structures into sequence-like forms (Behrouz & Hashemi, 2024; Wang et al., 2025b; Hu et al., 2025). However, node-order ambiguity and non-stationary evolution still hinder robust state-space modeling on dynamic graphs. Recent approaches like SpoT-Mamba (Choi et al., 2024) and SSM–based structure learning (Yuan et al., 2025) still fall short of robust performance. A more comprehensive review of related work is provided in Appendix I.

## 3. Preliminaries and Problem Formulation

**Notation.** In this work, we focus on discrete dynamic graph. A discrete dynamic graph with $T$ time steps is denoted as $\mathbf{DG} = \{\mathcal{G}_t\}_{t=1}^T$. The snapshot at time $t$ is defined as $\mathcal{G}_t = (\mathcal{V}_t, \mathcal{E}_t, \mathbf{X}_t)$, where $\mathcal{V}_t$ and $\mathcal{E}_t$ respectively denote the node and edge sets, and $N = |\mathcal{V}_t|$ is the number of nodes. The node feature matrix is $\mathbf{X}_t \in \mathbb{R}^{N \times d}$ ($d$ is the feature dimension), and the adjacency matrix is $\mathbf{A}_t \in \{0,1\}^{N \times N}$.

**Problem definition.** To test the performance of dynamic graph representation learning, we select the downstream task of dynamic link prediction. The objective is to forecast edge connectivity at time $T + 1$ given the historical graph sequence $\mathcal{G}_{1:T}$ and node features $\mathbf{X}_{T+1}$. Formally, the task is to learn a predictive mapping $f_\theta$ comprising an encoder $\varphi(\cdot)$ and a link predictor $\psi(\cdot)$. The encoder maps historical structural evolution and current features to latent embeddings $\mathbf{Z}_{T+1} = \varphi(\mathcal{G}_{1:T}, \mathbf{X}_{T+1})$, and the predictor estimates edges existence via the predicted adjacency matrix $\hat{\mathbf{Y}}_{T+1} = \psi(\mathbf{Z}_{T+1})$. The goal is to obtain robust dynamic

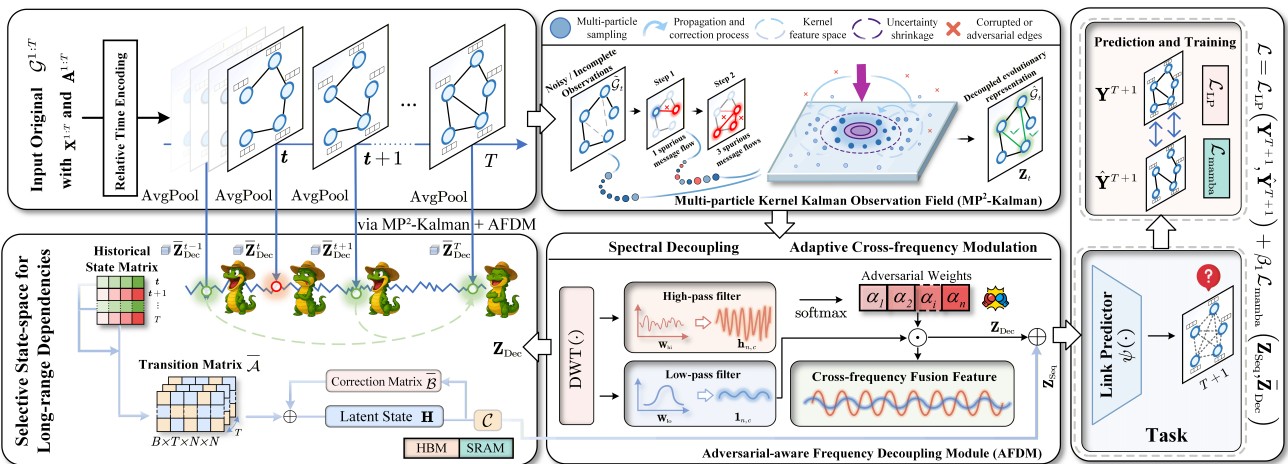

*Figure 2.* The framework of DeR-Mamba. (a) MP–K$^2$alman captures structural uncertainty by sampling latent evolution trajectories in multiple kernel subspaces and performing Kalman-style updates for state estimation and correction. (b) AFDM mitigates non-stationary spectral perturbations via wavelet-based spectral decoupling and adaptive cross-frequency modulation with adversarial weights. (c) Long-range dependencies selective modeling alleviates temporal redundancy arising from accumulated residual perturbations by discretizing historical adjacency representations and performing selective snapshot scanning to highlight critical temporal dependencies.

graph representations that support more accurate dynamic link prediction.

## 4. Methodology

In this section, we present DeR-Mamba (Figure 2), a framework that systematically decouples entangled multi-domain dynamics. We leverage the MP–K$^2$alman module to characterize structural uncertainty via kernel-based particle sampling for spatial decoupling. To address spectral instability, AFDM employs frequency-domain decoupling and dynamic cross-frequency modulation. Finally, we design a selective state-space mechanism to capture global dynamics, ensuring robust temporal decoupling.

### 4.1. Multi-particle Kernel Kalman Observation Field

To mitigate the negative effects of structural incompleteness, we propose MP–K$^2$alman. This component samples latent evolutionary paths within multiple kernel subspaces and employs Kalman-style updates to dynamically fuse confidence across diverse particle views. By approximating the posterior trajectory via Bayesian inference, MP–K$^2$alman explicitly decouples structural dynamics from perturbations. A formal Bayesian formulation of this process, along with the complete derivation of the observation model and MAP estimator, is provided in Appendix C.

**Kernel subspace sampling.** We sample $M$ groups of random projections in parallel, where each group corresponds to a structural hypothesis particle. For the $t$-th snapshot, the random projection matrix for the $m$-th particle ($m \in \{1, \ldots, M\}$) and the $h$-th attention head ($h \in \{1, \ldots, H\}$) is denoted as $\mathbf{R}^{(m,h)} \in \mathbb{R}^{d_\phi \times d}$. Grounded in Mercer's

theorem (Mercer, 1909), the Positive Random Features (PRF) (Choromanski et al., 2021) mapping projects input vectors into a randomized feature space that efficiently approximates the Softmax kernel. The element-wise exponential mapping for an input vector $u$ is defined as:

$$\phi_{\text{PRF}}^{(m,h)}(u) = \frac{1}{\sqrt{d_\phi}} \exp\left(\mathbf{R}^{(m,h)}u - \frac{1}{2}\|u\|_2^2 \mathbf{1}\right), \quad (1)$$

where $\mathbf{R}^{(m,h)}$ is drawn from a standard Gaussian distribution $\mathcal{N}(0, \sigma^{-2}\mathbf{I})$ and subsequently orthogonalized via the Orthogonal Random Features (ORF) (Yu et al., 2016) procedure to reduce variance. This mapping projects the observed graph $\tilde{\mathcal{G}}_t$ into multiple low-redundancy, complementary kernel subspaces, enhancing the discrimativeness of the latent distribution under perturbations.

**Multi-particle filtering system.** We construct a multi-particle filtering system by computing the Query ($\mathbf{Q}$) and Value ($\mathbf{V}$) matrices for each particle $m$ and head $h$:

$$\mathbf{Q}_t^{(m,h)} = \phi_q^{(m)}(\mathbf{W}_Q^{(m)}\mathbf{X}_t^{(m,h)}), \ \mathbf{V}_t^{(m,h)} = \mathbf{W}_V^{(m)}\mathbf{X}_t^{(m,h)}, \quad (2)$$

where $\mathbf{W}_{Q/V}^{(m)}$ are learnable linear projections. To estimate stable statistics, we compute the centered means across the $M$ particles:

$$\bar{\mathbf{Q}}_t^{(h)} = \frac{1}{M}\sum_{i=1}^{M}\phi_q^{(m)}(\mathbf{Q}_t^{(h)};\Pi_i), \ \tilde{\mathbf{Q}}_{t,i}^{(h)} = \phi_q^{(m)}(\mathbf{Q}_t^{(h)};\Pi_i) - \bar{\mathbf{Q}}_t^{(h)}, \quad (3)$$

$$\bar{\mathbf{V}}_t^{(h)} = \frac{1}{M}\sum_{i=1}^{M}\mathbf{V}_{t,i}^{(h)}, \ \tilde{\mathbf{V}}_{t,i}^{(h)} = \mathbf{V}_{t,i}^{(h)} - \bar{\mathbf{V}}_t^{(h)}, \quad (4)$$

where $\Pi_i = \mathbf{R}^{(i,h)}$ denotes the random projection parameters specific to the $i$-th particle, and $\phi(\cdot;\Pi_i)$ corresponds to

the explicit kernel mapping defined in Eq. (1). Subsequently, we estimate the element-wise variance and the Query-Value covariance within the random feature dimension $d_\phi$:

$$Var_{q,t}^{(h)} = \frac{1}{M} \sum_{i=1}^{M} \tilde{\mathbf{Q}}_{t,i}^{(h)} \odot \tilde{\mathbf{Q}}_{t,i}^{(h)} + \varepsilon, \quad (5)$$

$$Cov_{qv,t}^{(h)} = \frac{1}{M} \sum_{i=1}^{M} \tilde{\mathbf{Q}}_{t,i}^{(h)} \otimes \tilde{\mathbf{V}}_{t,i}^{(h)}, \quad (6)$$

where $Var_{q,t}^{(h)} \in \mathbb{R}^{N \times d_\phi}$ and $Cov_{qv,t}^{(h)} \in \mathbb{R}^{N \times d_\phi \times d}$ denote the variance and covariance tensors. We use $\odot$ for the element-wise product and $\otimes$ for the outer product across dimension $d_\phi$, with $\varepsilon = 10^{-6}$ ensuring numerical stability.

**Kalman-style update.** We derive the Kalman-style gain matrix $\mathbf{K}_{\text{gain},t}^{(h)}$ based on the Linear Minimum Mean Square Error (LMMSE) (Kay, 1993) principle. Notably, by constructing the gain using kernel feature statistics, we avoid the quadratic complexity of full attention maps ($\mathbf{QK}^\top$), reducing the complexity from $O(N^2)$ to $O(N)$:

$$\mathbf{K}_{\text{gain},t}^{(h)}(n,r,:) = \frac{Cov_{qv,t}^{(h)}(n,r,:)}{Var_{q,t}^{(h)}(n,r) + \varepsilon}, \quad (7)$$

where $\mathbf{K}_{\text{gain},t}^{(h)} \in \mathbb{R}^{N \times d_\phi \times d}$. The gain matrix adaptively balances the prior estimate and the observation, acting as a Bayesian confidence fusion mechanism. The final latent state $\tilde{\mathbf{Z}}_t^{(h)}$ is obtained via the LMMSE update using the mean observation $\bar{\mathbf{Q}}_t^{(h)}$:

$$\tilde{\mathbf{Z}}_t^{(h)}(n,:) = \sum_{r=1}^{d_\phi} \mathbf{K}_{\text{gain},t}^{(h)}(n,r,:) \bar{\mathbf{Q}}_t^{(h)}(n,r). \quad (8)$$

**Relational bias injection.** To further mitigate variance under high uncertainty conditions, we superimpose a structural relational bias onto the Kalman-updated representations. Let $\{\mathbf{A}_t^{(r)}\}_{r=1}^R$ denote the set of $r$-hop adjacency matrices, where each $\mathbf{A}_t^{(r)}$ represents the connectivity of neighbors at distance $r$. The $r$-th order degree-normalized adjacency operator $\hat{\mathbf{S}}_t^{(r)}$ and the final enhanced representation $\mathbf{Z}_t^{(h)}$ are defined as:

$$\hat{\mathbf{S}}_t^{(r)} = \hat{\mathbf{D}}_t^{(r)-1/2} \mathbf{A}_t^{(r)} \hat{\mathbf{D}}_t^{(r)-1/2}, \quad (9)$$

$$\mathbf{Z}_t^{(h)} = \tilde{\mathbf{Z}}_t^{(h)} + \sum_{r=1}^{R} \hat{\mathbf{S}}_t^{(r)} \bar{\mathbf{V}}_t^{(h)} \sigma(b_r^{(h)}), \quad (10)$$

where $\mathbf{A}_t^{(r)}, \hat{\mathbf{S}}_t^{(r)} \in \mathbb{R}^{N \times N}$. $\hat{\mathbf{D}}_t^{(r)}$ and $b_r^{(h)}$ denote the corresponding degree matrix and the learnable scalar weight for the $r$-th order neighbor at the $h$-th head, respectively. $\sigma(\cdot)$ is the sigmoid function. This step smooths the contributions of multi-hop neighbors, resulting in a robust structural encoding. We theoretically prove that MP-K²alman maximizes the structural information gain and guarantees asymptotic optimality under uncertainty, as detailed in Appendix B.1.

## 4.2. Adversarial-aware Frequency Decoupling Module

Temporal node signals in dynamic graphs inherently consist of stationary low-frequency components, which represent the system's long-term evolutionary patterns, and fluctuating high-frequency components, which often encapsulate local abrupt changes. However, in adversarial settings, the latter is highly susceptible to noise contamination and attacks. If left unchecked, these misleading spectral signals accumulate and diffuse during temporal propagation, causing severe representation distortion. To mitigate this issue, we introduce AFDM, which employs a differentiable wavelet-based decoupling mechanism to isolate non-stationary fluctuations and an adaptive cross-frequency modulation scheme to purify the spectral signals.

**Differentiable frequency decoupling.** To achieve fine-grained spectral separation while maintaining differentiability, we employ a non-decimated 1D Discrete Wavelet Transform (DWT) along the temporal dimension. We derive low-pass and high-pass analysis filters, denoted as $\omega_{\text{lo}}$ and $\omega_{\text{hi}}$, respectively. To facilitate unified matrix operations within the neural network, we transform these filters into two sets of sparse Toeplitz matrices: $\mathbf{W}_{\text{lo}}, \mathbf{W}_{\text{hi}} \in \mathbb{R}^{T \times T}$. The entries of these matrices are explicitly defined as:

$$[\mathbf{W}_{\text{lo}}]_{t,\tau} = \begin{cases} \omega_{\text{lo}}[\tau - (t - k_0)], & 0 \le \tau - (t - k_0) < k \\ 0, & \text{otherwise,} \end{cases} \quad (11)$$

$$[\mathbf{W}_{\text{hi}}]_{t,\tau} = \begin{cases} \omega_{\text{hi}}[\tau - (t - k_0)], & 0 \le \tau - (t - k_0) < k \\ 0, & \text{otherwise,} \end{cases} \quad (12)$$

where $k$ denotes the filter length, $k_0 = \lfloor k/2 \rfloor$ is the alignment shift, $t$ represents the sampling position, and $\tau$ is the input snapshot index. Through this construction, the $t$-th row of $\mathbf{W}_{\text{lo}}$ (or $\mathbf{W}_{\text{hi}}$) effectively acts as a weight template shifted to position $t$, making the matrix-vector multiplication equivalent to a 1D convolution over a local window centered at $t$. Consequently, the wavelet decomposition for a node $n$ at channel $c$ can be formulated as:

$$\mathbf{l}_{n,c} = \mathbf{Z}_{n,c} \mathbf{W}_{\text{lo}}^\top, \quad \mathbf{h}_{n,c} = \mathbf{Z}_{n,c} \mathbf{W}_{\text{hi}}^\top, \quad (13)$$

where $\mathbf{l}_{n,c}, \mathbf{h}_{n,c} \in \mathbb{R}^T$ represent the low-frequency and high-frequency components, respectively. Stacking all nodes yields the global spectral representations $\mathbf{L}, \mathbf{H} \in \mathbb{R}^{N \times T \times C}$. Since the DWT is a bounded linear map satisfying the 1-Lipschitz condition, the gradients can propagate stably:

$$\frac{\partial \mathcal{L}}{\partial \mathbf{x}_{n,c}} = \frac{\partial \mathcal{L}}{\partial \mathbf{l}_{n,c}} \mathbf{W}_{\text{lo}} + \frac{\partial \mathcal{L}}{\partial \mathbf{h}_{n,c}} \mathbf{W}_{\text{hi}}. \quad (14)$$

**Adversarial-aware weight generation.** Instead of employing heuristic gating functions, we formulate the generation of the adversarial weights $\boldsymbol{\alpha}_{n,c}$ for the $c$-th feature of

node $n$ as a constrained optimization problem. Our objective is to maximize the alignment with the high-frequency perturbation signals while maintaining a high-entropy distribution to prevent overfitting to local noise spikes. This is formalized as:

$$\boldsymbol{\alpha}_{n,c} = \underset{\mathbf{a} \in \mathcal{U}^{T-1}}{\arg \max} \left( \langle \mathbf{a}, \mathbf{H}[n, :, c] \rangle + \lambda \mathcal{H}(\mathbf{a}) \right), \qquad (15)$$

where $\mathcal{U}^{T-1}$ denotes the probability simplex, $\mathcal{H}(\mathbf{a}) = -\sum_t a_t \log a_t$ represents the Shannon entropy, and $\lambda$ is a Lagrangian multiplier controlling the regularization strength. The closed-form solution to this convex objective is given by the Gibbs distribution (LeCun et al., 2006):

$$\alpha_{n,c}^{(t)} = \frac{\exp \left( \frac{1}{\lambda} \mathbf{H}[n, t, c] \right)}{\sum_{k=1}^{T} \exp \left( \frac{1}{\lambda} \mathbf{H}[n, k, c] \right)}. \qquad (16)$$

Finally, the frequency modulation is executed via element-wise gating, acting as a differentiable spectral filter:

$$\mathbf{y}_{n,c} = \mathbf{l}_{n,c} \odot \boldsymbol{\alpha}_{n,c} \in \mathbb{R}^T, \qquad (17)$$

where $\odot$ denotes element-wise multiplication. By framing the weight generation as an entropy-regularized problem, AFDM adaptively balances the pursuit of high alignment to capture salient abrupt changes with the maintenance of high entropy to ensure temporal continuity. Finally, the output of AFDM is obtained by integrating the modulated sequences across the temporal dimension:

$$\mathbf{Z}_{\text{Dec}} = \text{AFDM}(\mathbf{Z}) \in \mathbb{R}^{N \times T \times C}, \ \mathbf{Z}_{\text{Dec}}[n, :, c] = \mathbf{y}_{n,c}. \ (18)$$

By dynamically adapting $\boldsymbol{\alpha}$, AFDM can suppress weights during disturbance bursts and reinforces low-frequency signals in stationary intervals, achieving robust spectral noise decoupling. We theoretically prove that AFDM can effectively decouple high-frequency noise from signal propagation to ensure spectral stability, as detailed in Appendix B.2.

### 4.3. Selective State-space for Long-range Dependencies

Though MP–K$^2$alman and AFDM suppress snapshot-level distortions, the accumulation of residual perturbations across the temporal domain induces temporal redundancy. To address this issue, we introduce a selective state-space mechanism that preserves informative and key states while suppressing residual perturbation accumulation, enabling robust long-range dynamic modeling.

**State-space based dynamic graph modeling.** Existing methods face a dilemma: capturing global dependencies incurs prohibitive spatial costs, while modeling only local structures accumulates residual perturbations, forming temporal redundancy. To resolve this, we model the dynamic graph as a self-contained system driven by State Space Models (SSMs), enabling selective long-range modeling without

increasing spatial complexity. Specifically, SSMs capture evolution via a transition matrix $\mathcal{A} \in \mathbb{R}^{n \times n}$ and projection matrices $\mathcal{B} \in \mathbb{R}^{n \times 1}, \mathcal{C} \in \mathbb{R}^{1 \times n}$. We define the Historical State Matrix as $\{\mathbf{h}(t)\}_{t=1}^{T} \in \mathbb{R}^{T \times n \times l}$. Given a continuous input sequence $\mathbf{x}(t) \in \mathbb{R}^l$, the SSM updates the latent state $\mathbf{h}(t) \in \mathbb{R}^{n \times l}$ and produces the output $\mathbf{y}(t) \in \mathbb{R}^l$, *i.e.*,

$$\mathbf{h}'(t) = \mathcal{A}\mathbf{h}(t) + \mathcal{B}\mathbf{x}(t), \quad \mathbf{y}(t) = \mathcal{C}\mathbf{h}(t), \quad (19)$$

where $\mathcal{A}$ governs the global temporal evolution of the latent states, $\mathcal{B}$ models the driving influence of the input, and $\mathcal{C}$ maps the latent state to the output space.

To effectively deploy the continuous system in Eq. (19) within deep learning frameworks, discretization is essential, and two key challenges must be addressed:

- **Q1:** How can the SSM be equipped with attention-like selectivity to perceive snapshot-specific states and identify salient global dependencies, while avoiding the quadratic complexity inherent to self-attention?
- **Q2:** How can local Hidden Markov constraints be overcome to suppress the uncontrolled accumulation of redundant patterns across long temporal horizons?

**Selective discretizing with redundancy-suppressed state updates.** To resolve these challenges, a Dynamic Graph Redundancy-Aware Scanning (DGRS) mechanism is developed. For Q1, DGRS discretizes the dynamic system parameters $(\mathcal{A}, \mathcal{B}, \mathcal{C}, \text{etc.})$, thereby parameterizing the time-varying discretization step size $\Delta_t$ (defined in Eq. (21)) and enabling linear-complexity control over which snapshots are incorporated into the hidden state. For Q2, inter-graph structures $\hat{\mathbf{A}}_{\text{inter}}^{1:t}$ (where $u \in \mathcal{V}^t$ and $v \in \mathcal{V}^{t-1}$) are used to modulate the discretization step size $\Delta_t$ based on structural variation, allowing local structural changes to guide temporal state updates and strengthen long-range dependency modeling.

Specifically, we extend the decoupled representation $\mathbf{Z}_{\text{Dec}}$ in Eq. (18) to $\mathbb{R}^{B \times T \times N \times C}$ to accommodate batch processing. A sequence input $\bar{\mathbf{Z}}_{\text{Dec}} \in \mathbb{R}^{B \times T \times N}$ is then obtained by applying average pooling over feature dimensions. The transition matrix $\mathcal{A} \in \mathbb{R}^{N \times D}$ is randomly initialized, while $\mathcal{B}$ and $\mathcal{C}$ are step-wise parameterized as:

$$\mathcal{B}, \mathcal{C} \in \mathbb{R}^{B \times T \times D} \leftarrow \text{Linear}_D \left( \bar{\mathbf{Z}}_{\text{Dec}} \right), \qquad (20)$$

where $\mathcal{B}$ regulates how external inputs drive latent-state evolution, and $\mathcal{C}$ extracts key structural change signals to suppress temporal redundancy. A snapshot-wise time-step parameter $\Delta \in \mathbb{R}^{B \times T \times N}$ is further introduced and combined with inter-graph structures $\hat{\mathbf{A}}_{\text{inter}}^{1:T}$ to perform time-wise discretization:

$$\Delta \in \mathbb{R}^{B \times T \times N} \leftarrow \text{softplus} \left( \text{Linear}_N \left( \bar{\mathbf{Z}}_{\text{Dec}} \right) \right), \qquad (21)$$

$$\overline{\Delta} \in \mathbb{R}^{B \times T \times N} \leftarrow \text{unsqueeze}_N(\Delta) \cdot \mathbf{W} \hat{\mathbf{A}}_{\text{inter}}^{1:T}, \qquad (22)$$

where softplus$(\cdot)$ ensures a non-negative time-step parameter $\Delta$. Following the zero-order hold principle, the continuous structural dynamics are discretized, and a Taylor expansion yields the discrete temporal operators $\overline{\mathcal{A}}$ and $\overline{\mathcal{B}}$:

$$\overline{\mathcal{A}} = \exp(\overline{\Delta}\mathcal{A}) = \mathbf{I} + \overline{\Delta}\mathcal{A} + \frac{1}{2}(\overline{\Delta}\mathcal{A})^2 + \frac{1}{6}(\overline{\Delta}\mathcal{A})^3 + \cdots, \tag{23}$$

$$\overline{\mathcal{B}} = (\overline{\Delta}\mathcal{A})^{-1}\left(\exp(\overline{\Delta}\mathcal{A}) - \mathbf{I}\right)\overline{\Delta}\mathcal{B}$$

$$= \left[\sum_{k=0}^{\infty} \frac{1}{(k+1)!}(\overline{\Delta}\mathcal{A})^k\right]\overline{\Delta}\mathcal{B}, \tag{24}$$

where $\overline{\mathcal{A}}, \overline{\mathcal{B}} \in \mathbb{R}^{B \times T \times N \times D}$. The final output of the dynamic graph system is:

$$\mathcal{S}_t = \overline{\mathcal{A}}\mathcal{S}_{t-1} + \overline{\mathcal{B}}(\overline{\mathbf{Z}}_{\text{Dec}})_t, \quad (\mathbf{Z}_{\text{Seq}})_t = \mathcal{C}\mathcal{S}_t, \tag{25}$$

where $\mathcal{S} \in \mathbb{R}^{B \times T \times N \times D}$ denotes the latent state. The procedure of DGRS is provided in Algorithm 2 (see Appendix A). The resulting redundancy-aware representation $\mathbf{Z}_{\text{Seq}} \in \mathbb{R}^{B \times T \times N}$ selectively captures global long-range dependencies and discriminative temporal information during evolution, effectively mitigating temporal redundancy caused by residual disturbances between snapshots. Finally, the ultimate robust representation $\hat{\mathbf{Z}}$ is given by:

$$\hat{\mathbf{Z}} = \mathbf{Z}_{\text{Dec}} + \eta \cdot \text{unsqueeze}_{D_0}(\mathbf{Z}_{\text{Seq}}), \tag{26}$$

where the coefficient $\eta$ controls how strongly the long-range temporal component contributes to the final representation. The computation in Eq. (25) enables linear-time scaling, and the implementation details are provided in Appendix J.1.

### 4.4. Optimization and Complexity Analysis

The overall learning objective of DeR-Mamba is to minimize a composite loss function that balances accurate dynamic link prediction with temporal redundancy suppression. Formally, the total objective $\mathcal{L}$ is defined as:

$$\mathcal{L} = \mathcal{L}_{\text{LP}}\left(\mathbf{Y}^{T+1}, \hat{\mathbf{Y}}^{T+1}\right) + \beta_1 \mathcal{L}_{\text{mamba}}\left(\mathbf{Z}_{\text{Seq}}, \overline{\mathbf{Z}}_{\text{Dec}}\right), \tag{27}$$

where $\hat{\mathbf{Y}}^{T+1} = \psi(\hat{\mathbf{Z}})$ denotes the final prediction, and $\mathcal{L}_{\text{LP}}$ is the cross-entropy loss for the future link prediction task. $\mathcal{L}_{\text{mamba}}$ acts as a redundancy-suppression regularizer, governed by Lagrangian hyperparameter $\beta_1$, formulated as:

$$\mathcal{L}_{\text{mamba}}\left(\mathbf{Z}_{\text{Seq}}, \overline{\mathbf{Z}}_{\text{Dec}}\right) = \frac{1}{T}\sum_{t=1}^{T}\underbrace{\mathcal{H}(\text{softmax}(\mathbf{Z}_{\text{Seq},t}))}_{\text{Entropy}} \tag{28}$$

$$+ \frac{1}{T}\sum_{t=1}^{T}\beta_2 \cdot \underbrace{\mathcal{D}_{\text{KL}}\left(\text{softmax}(\mathbf{Z}_{\text{Seq},t}) \,\middle\|\, \text{softmax}(\overline{\mathbf{Z}}_{\text{Dec},t})\right)}_{\text{KL divergence}},$$

where $\beta_2$ balances the entropy term $\mathcal{H}(p) = -\sum p_i \log p_i$, which promotes discriminative signal learning, and the KL divergence $\mathcal{D}_{\text{KL}}(p|q) = \sum p_i \log \frac{p_i}{q_i}$, which enforces distributional consistency to constrain deviations accumulated during state evolution. The complete training pipeline is provided in Algorithm 1 (see Appendix A).

## 5. Experiments

In this section, we conduct extensive experiments on real-world dynamic graph datasets to evaluate the effectiveness and robustness of DeR-Mamba under multi-source adversarial perturbations. Implementation details and additional results are provided in Appendix F.

### 5.1. Experimental Setup

**Datasets.** We select real-world dynamic graph datasets covering temporal ranges of 16 years, 24 months, and 30 days, forming a multi-scale evaluation setting. ① **COLLAB** (Tang et al., 2012) has 16 yearly snapshots of citation and collaboration relationships among researchers. ② **Yelp** (Sankar et al., 2020) provides 24 monthly snapshots of user–business interactions, while ③ **ACT** (Kumar et al., 2019) contains 30 daily snapshots of user-user interactions.

**Baselines.** To evaluate DeR-Mamba, we compare it with state-of-the-art (SOTA) baselines, including ① **Static GNNs:** GAE (Kipf & Welling, 2016), VGAE (Kipf & Welling, 2016), GAT (Velickovic et al., 2018). ② **Dynamic GNNs:** GCRN (Seo et al., 2018), EvolveGCN (Pareja et al., 2020), DySAT (Sankar et al., 2020), SpoT-Mamba (Choi et al., 2024). ③ **Dynamic Graph Structure Learning (DGSL):** RDGSL (Zhang et al., 2023b), TGSL (Zhang et al., 2023a), and ④ **Robust DGNNs:** RGCN (Zhu et al., 2019), WinGNN (Zhu et al., 2023), DGIB (Yuan et al., 2024), DG-Mamba (Yuan et al., 2025).

**Adversarial attack settings.** To systematically evaluate the robustness of DeR-Mamba under different adversarial conditions, we consider two representative attack scenarios:

- **Non-targeted**: We apply random perturbations to graph structures and node features. ① **Structure attack**: Randomly adding or removing edges to simulate incomplete structural observations and corrupted propagation paths. ② **Feature attack**: Injecting Gaussian noise $\gamma \cdot \theta \cdot \epsilon$ into node features to mimic high-frequency and instantaneous disturbances, where $\gamma \in \{0.5, 1.0, 1.5\}$ controls the noise level, $\theta$ denotes amplitude, and $\epsilon \sim \mathcal{N}(0, \mathbf{I})$.

- **Targeted**: We employ the NETTACK attack (Zügner et al., 2018) to modify node neighborhoods or node features in a targeted manner. ① **Evasion attack**: Perturbations are injected only during testing after training on clean data. ② **Poisoning attack**: Perturbations are applied before both training and testing. GAT (Velickovic et al., 2018) is used as the surrogate model to guide the attack, with perturbation counts $n_p \in \{1, 2, 3\}$.

**Hyperparameter settings.** We follow standard configurations from the original baseline papers. All baselines use two GNN layers, while DeR-Mamba adopts a single layer. The latent dimension is 128. The maximum number of training epochs is 2000. Validation-based grid search is used, and detailed settings are provided in Appendix G.

*Table 1.* AUC scores (% ± standard deviation over five trials) of dynamic link prediction on real-world datasets under non-targeted attacks. The highest scores are highlighted in **bold**, and the runner-ups are underlined.

| Dataset | COLLAB | | | | | Yelp | | | | | ACT | | | | |
|---|---|---|---|---|---|---|---|---|---|---|---|---|---|---|---|
| Model | Original | Structure attack | Feature attack | | | Original | Structure attack | Feature attack | | | Original | Structure attack | Feature attack | | |
| | | | $\gamma$=0.5 | $\gamma$=1.0 | $\gamma$=1.5 | | | $\gamma$=0.5 | $\gamma$=1.0 | $\gamma$=1.5 | | | $\gamma$=0.5 | $\gamma$=1.0 | $\gamma$=1.5 |
| GAE | $77.15_{\pm0.5}$ | $74.04_{\pm0.8}$ | $50.59_{\pm0.8}$ | $44.66_{\pm0.8}$ | $43.12_{\pm0.8}$ | $70.67_{\pm1.1}$ | $64.45_{\pm5.2}$ | $51.05_{\pm0.6}$ | $45.41_{\pm0.6}$ | $41.56_{\pm0.9}$ | $72.31_{\pm0.5}$ | $60.27_{\pm0.4}$ | $56.56_{\pm0.5}$ | $52.52_{\pm0.6}$ | $50.36_{\pm0.9}$ |
| VGAE | $86.47_{\pm0.0}$ | $74.95_{\pm1.2}$ | $56.75_{\pm0.6}$ | $50.39_{\pm0.7}$ | $48.68_{\pm0.7}$ | $76.54_{\pm0.5}$ | $65.33_{\pm1.4}$ | $55.53_{\pm0.7}$ | $49.88_{\pm0.6}$ | $45.08_{\pm0.6}$ | $79.18_{\pm0.5}$ | $66.29_{\pm1.3}$ | $60.67_{\pm0.7}$ | $57.39_{\pm0.8}$ | $55.27_{\pm1.0}$ |
| GAT | $88.26_{\pm0.4}$ | $77.29_{\pm1.8}$ | $58.13_{\pm0.9}$ | $51.41_{\pm0.9}$ | $49.77_{\pm0.9}$ | $77.93_{\pm0.1}$ | $69.35_{\pm1.6}$ | $56.72_{\pm0.3}$ | $52.51_{\pm0.5}$ | $46.21_{\pm0.5}$ | $85.07_{\pm0.3}$ | $66.05_{\pm1.2}$ | $66.05_{\pm0.4}$ | $61.85_{\pm0.3}$ | $59.05_{\pm0.3}$ |
| GCRN | $82.78_{\pm0.5}$ | $69.72_{\pm0.9}$ | $54.07_{\pm0.9}$ | $47.78_{\pm0.8}$ | $46.18_{\pm0.9}$ | $68.59_{\pm1.0}$ | $54.68_{\pm7.6}$ | $52.68_{\pm0.6}$ | $46.85_{\pm0.6}$ | $40.45_{\pm0.6}$ | $76.28_{\pm0.5}$ | $64.35_{\pm1.2}$ | $59.48_{\pm0.7}$ | $54.16_{\pm0.6}$ | $53.88_{\pm0.7}$ |
| EvolveGCN | $86.62_{\pm1.0}$ | $76.15_{\pm0.9}$ | $56.82_{\pm1.2}$ | $50.33_{\pm1.0}$ | $48.55_{\pm1.0}$ | $78.21_{\pm0.0}$ | $53.82_{\pm2.0}$ | $57.91_{\pm0.5}$ | $51.82_{\pm0.3}$ | $45.32_{\pm0.4}$ | $74.55_{\pm0.3}$ | $63.17_{\pm1.0}$ | $61.02_{\pm0.5}$ | $53.34_{\pm0.5}$ | $51.62_{\pm0.7}$ |
| DySAT | $88.77_{\pm0.2}$ | $76.59_{\pm0.2}$ | $58.28_{\pm0.3}$ | $51.52_{\pm0.3}$ | $49.32_{\pm0.5}$ | $78.87_{\pm0.6}$ | $66.09_{\pm1.4}$ | $58.46_{\pm0.4}$ | $52.33_{\pm0.7}$ | $46.24_{\pm0.7}$ | $78.52_{\pm0.4}$ | $66.55_{\pm1.2}$ | $61.94_{\pm0.8}$ | $56.98_{\pm0.6}$ | $54.14_{\pm0.7}$ |
| SpoT-Mamba | $84.34_{\pm0.4}$ | $74.39_{\pm0.2}$ | $54.76_{\pm0.8}$ | $48.64_{\pm0.9}$ | $47.25_{\pm0.7}$ | $77.01_{\pm1.0}$ | $60.56_{\pm1.2}$ | $54.72_{\pm0.8}$ | $50.11_{\pm0.8}$ | $44.95_{\pm0.8}$ | $73.29_{\pm1.0}$ | $61.27_{\pm0.9}$ | $59.92_{\pm0.7}$ | $52.19_{\pm0.8}$ | $51.33_{\pm0.9}$ |
| RDGSL | $82.29_{\pm0.5}$ | $71.36_{\pm0.9}$ | $52.33_{\pm0.6}$ | $48.50_{\pm0.7}$ | $45.21_{\pm0.6}$ | $75.92_{\pm0.6}$ | $58.30_{\pm0.9}$ | $52.29_{\pm0.5}$ | $48.66_{\pm0.4}$ | $44.59_{\pm0.5}$ | $73.15_{\pm0.6}$ | $62.45_{\pm1.0}$ | $60.14_{\pm0.6}$ | $53.05_{\pm0.5}$ | $51.07_{\pm0.6}$ |
| TGSL | $84.09_{\pm0.5}$ | $73.66_{\pm1.0}$ | $55.29_{\pm0.4}$ | $51.34_{\pm0.4}$ | $50.28_{\pm0.3}$ | $76.55_{\pm0.4}$ | $73.29_{\pm1.1}$ | $60.21_{\pm0.3}$ | $51.01_{\pm0.3}$ | $49.87_{\pm0.4}$ | $80.53_{\pm0.5}$ | $70.32_{\pm0.9}$ | $67.19_{\pm0.4}$ | $60.27_{\pm0.5}$ | $58.39_{\pm0.5}$ |
| RGCN | $88.21_{\pm0.1}$ | $78.66_{\pm0.7}$ | $61.29_{\pm0.5}$ | $54.29_{\pm0.6}$ | $52.99_{\pm0.6}$ | $77.28_{\pm0.3}$ | $74.29_{\pm0.4}$ | $59.72_{\pm0.3}$ | $52.88_{\pm0.3}$ | $50.40_{\pm0.2}$ | $87.22_{\pm0.2}$ | $82.66_{\pm0.4}$ | $68.51_{\pm0.2}$ | $62.67_{\pm0.2}$ | $61.31_{\pm0.2}$ |
| WinGNN | $90.33_{\pm0.1}$ | $82.34_{\pm0.6}$ | $64.69_{\pm0.9}$ | $56.87_{\pm1.1}$ | $54.44_{\pm0.6}$ | $76.46_{\pm1.0}$ | $74.59_{\pm0.8}$ | $60.45_{\pm0.4}$ | $55.80_{\pm1.0}$ | $52.73_{\pm0.8}$ | $90.12_{\pm0.4}$ | $85.36_{\pm0.4}$ | $71.60_{\pm0.9}$ | $65.40_{\pm0.3}$ | $63.32_{\pm0.8}$ |
| DGIB-Bern | $92.17_{\pm0.2}$ | $83.58_{\pm0.1}$ | $63.54_{\pm0.9}$ | $56.92_{\pm1.0}$ | $56.24_{\pm1.0}$ | $76.88_{\pm0.2}$ | $75.61_{\pm0.0}$ | $\underline{63.91}_{\pm0.9}$ | $\mathbf{59.28}_{\pm0.9}$ | $54.77_{\pm1.0}$ | $94.49_{\pm0.2}$ | $87.75_{\pm0.1}$ | $73.05_{\pm0.9}$ | $68.49_{\pm0.9}$ | $66.27_{\pm0.9}$ |
| DGIB-Cat | $92.68_{\pm0.1}$ | $84.16_{\pm0.1}$ | $63.99_{\pm0.5}$ | $57.76_{\pm0.8}$ | $55.63_{\pm1.0}$ | $79.53_{\pm0.2}$ | $77.72_{\pm0.1}$ | $61.42_{\pm0.9}$ | $55.12_{\pm0.7}$ | $51.90_{\pm0.9}$ | $94.89_{\pm0.2}$ | $88.27_{\pm0.2}$ | $73.92_{\pm0.8}$ | $68.88_{\pm0.9}$ | $65.99_{\pm0.7}$ |
| DG-Mamba | $93.60_{\pm0.3}$ | $92.60_{\pm0.3}$ | $68.53_{\pm1.5}$ | $60.88_{\pm1.0}$ | $56.95_{\pm0.8}$ | $81.54_{\pm0.6}$ | $77.40_{\pm0.7}$ | $61.82_{\pm0.9}$ | $57.42_{\pm0.6}$ | $55.97_{\pm1.2}$ | $96.67_{\pm0.3}$ | $96.14_{\pm0.3}$ | $79.36_{\pm0.8}$ | $73.76_{\pm0.7}$ | $70.21_{\pm0.7}$ |
| DeR-Mamba | $\mathbf{94.83}_{\pm0.1}$ | $\mathbf{93.74}_{\pm0.2}$ | $\mathbf{70.16}_{\pm0.4}$ | $\mathbf{61.99}_{\pm0.2}$ | $\mathbf{57.50}_{\pm0.3}$ | $\mathbf{84.66}_{\pm0.2}$ | $\mathbf{79.28}_{\pm0.3}$ | $61.60_{\pm0.2}$ | $\underline{58.32}_{\pm0.2}$ | $\mathbf{56.94}_{\pm0.5}$ | $\mathbf{97.17}_{\pm0.2}$ | $\mathbf{96.60}_{\pm0.2}$ | $\mathbf{80.78}_{\pm0.4}$ | $\mathbf{75.83}_{\pm0.3}$ | $\mathbf{73.48}_{\pm0.2}$ |

*Table 2.* AUC scores (% ± standard deviation over five trials) of dynamic link prediction on real-world datasets under targeted attacks. The highest scores are highlighted in **bold**, and the runner-ups are underlined.

| Dataset | Model | Original | Evasion attack | | | | Poisoning attack | | | |
|---|---|---|---|---|---|---|---|---|---|---|
| | | | $n_p$=1($\Delta\%_\downarrow$) | $n_p$=2($\Delta\%_\downarrow$) | $n_p$=3($\Delta\%_\downarrow$) | Avg. $\Delta\%_\downarrow$ | $n_p$=1($\Delta\%_\downarrow$) | $n_p$=2($\Delta\%_\downarrow$) | $n_p$=3($\Delta\%_\downarrow$) | Avg. $\Delta\%_\downarrow$ |
| COLLAB | GAT | $88.26_{\pm0.4}$ | $76.21_{\pm0.1}$(13.7) | $66.56_{\pm0.1}$(24.6) | $57.92_{\pm0.1}$(34.4) | 24.2 | $66.59_{\pm0.5}$(24.6) | $55.31_{\pm0.6}$(37.3) | $51.34_{\pm0.7}$(41.8) | 34.6 |
| | DySAT | $88.77_{\pm0.2}$ | $77.91_{\pm0.1}$(12.2) | $68.22_{\pm0.1}$(23.1) | $58.82_{\pm0.1}$(33.7) | 23.0 | $69.02_{\pm0.3}$(22.2) | $57.62_{\pm0.3}$(35.1) | $52.76_{\pm0.3}$(40.6) | 32.6 |
| | SpoT-Mamba | $84.34_{\pm0.4}$ | $71.45_{\pm0.2}$(15.3) | $65.88_{\pm0.2}$(21.9) | $52.14_{\pm0.3}$(38.2) | 25.1 | $66.45_{\pm0.5}$(21.2) | $55.36_{\pm0.9}$(34.4) | $53.17_{\pm0.6}$(37.0) | 30.8 |
| | TGSL | $84.09_{\pm0.5}$ | $72.09_{\pm0.3}$(14.3) | $65.30_{\pm0.2}$(22.3) | $52.09_{\pm0.3}$(38.1) | 24.9 | $66.57_{\pm0.2}$(20.8) | $54.21_{\pm0.2}$(35.5) | $55.36_{\pm0.3}$(34.2) | 30.2 |
| | WinGNN | $90.33_{\pm0.1}$ | $79.35_{\pm0.2}$(12.2) | $68.24_{\pm0.1}$(24.5) | $61.07_{\pm0.3}$(32.4) | 23.0 | $71.53_{\pm0.8}$(20.8) | $61.57_{\pm1.1}$(31.8) | $55.27_{\pm1.0}$(38.8) | 30.5 |
| | DGIB-Cat | $92.68_{\pm0.1}$ | $81.29_{\pm0.0}$(12.3) | $71.32_{\pm0.1}$(23.0) | $62.03_{\pm0.1}$(33.1) | 22.8 | $72.55_{\pm0.2}$(21.7) | $60.99_{\pm0.3}$(34.2) | $55.62_{\pm0.4}$(40.0) | 32.0 |
| | DG-Mamba | $93.60_{\pm0.3}$ | $81.78_{\pm0.6}$(12.6) | $80.87_{\pm0.6}$(13.6) | $68.75_{\pm1.3}$(26.5) | 17.6 | $79.48_{\pm0.2}$(15.1) | $67.45_{\pm0.1}$(27.9) | $64.99_{\pm0.6}$(30.6) | 24.5 |
| | DeR-Mamba | $\mathbf{94.83}_{\pm0.1}$ | $\mathbf{82.77}_{\pm0.6}$(12.7) | $\mathbf{81.13}_{\pm0.7}$(14.4) | $\mathbf{69.42}_{\pm1.0}$(26.8) | $\underline{18.0}$ | $\mathbf{82.53}_{\pm0.3}$(12.3) | $\mathbf{81.01}_{\pm0.4}$(14.6) | $\mathbf{68.38}_{\pm0.4}$(27.9) | **18.3** |
| Yelp | GAT | $77.93_{\pm0.1}$ | $67.96_{\pm0.1}$(12.8) | $59.47_{\pm0.1}$(23.7) | $50.27_{\pm0.1}$(35.5) | 24.0 | $65.34_{\pm0.5}$(16.2) | $54.51_{\pm0.2}$(30.1) | $50.24_{\pm0.6}$(35.5) | 27.2 |
| | DySAT | $78.87_{\pm0.6}$ | $69.77_{\pm0.1}$(11.5) | $60.66_{\pm0.1}$(23.1) | $52.16_{\pm0.1}$(33.9) | 22.8 | $66.87_{\pm0.6}$(15.2) | $56.31_{\pm0.3}$(28.6) | $50.44_{\pm0.6}$(36.0) | 26.6 |
| | SpoT-Mamba | $77.01_{\pm1.0}$ | $65.25_{\pm0.2}$(15.3) | $54.33_{\pm0.2}$(29.5) | $47.75_{\pm0.2}$(38.0) | 27.6 | $64.39_{\pm1.0}$(16.4) | $55.21_{\pm0.9}$(28.3) | $50.33_{\pm1.1}$(34.6) | 26.4 |
| | TGSL | $76.55_{\pm0.4}$ | $65.03_{\pm0.3}$(15.0) | $54.29_{\pm0.3}$(29.1) | $47.81_{\pm0.3}$(37.5) | 27.2 | $64.08_{\pm0.8}$(16.3) | $56.27_{\pm0.6}$(26.5) | $51.20_{\pm0.8}$(33.1) | 25.3 |
| | WinGNN | $76.46_{\pm1.0}$ | $66.25_{\pm1.0}$(13.4) | $60.22_{\pm0.9}$(21.2) | $51.38_{\pm0.8}$(32.8) | 22.5 | $67.88_{\pm0.9}$(11.2) | $56.36_{\pm0.9}$(26.3) | $52.74_{\pm1.0}$(31.0) | 22.8 |
| | DGIB-Cat | $79.53_{\pm0.2}$ | $70.17_{\pm0.1}$(11.8) | $62.25_{\pm0.1}$(21.7) | $52.69_{\pm0.1}$(33.7) | 22.4 | $67.38_{\pm0.3}$(15.3) | $57.02_{\pm0.2}$(28.3) | $51.39_{\pm0.2}$(35.4) | 26.3 |
| | DG-Mamba | $81.54_{\pm0.6}$ | $70.88_{\pm0.3}$(13.1) | $69.77_{\pm0.5}$(14.4) | $49.93_{\pm0.6}$(38.8) | $\underline{22.1}$ | $73.10_{\pm0.4}$(10.4) | $64.65_{\pm0.1}$(20.7) | $54.67_{\pm0.3}$(33.0) | $\underline{21.3}$ |
| | DeR-Mamba | $\mathbf{84.46}_{\pm0.2}$ | $\mathbf{77.57}_{\pm0.6}$(8.2) | $\mathbf{76.33}_{\pm0.7}$(9.6) | $\mathbf{64.28}_{\pm1.0}$(23.9) | **13.9** | $\mathbf{77.84}_{\pm0.5}$(7.8) | $\mathbf{76.67}_{\pm0.3}$(9.2) | $\mathbf{64.85}_{\pm0.3}$(23.2) | **13.4** |
| ACT | GAT | $85.07_{\pm0.3}$ | $75.14_{\pm0.1}$(11.7) | $67.25_{\pm0.1}$(20.9) | $59.75_{\pm0.1}$(29.8) | 20.8 | $71.26_{\pm0.9}$(16.2) | $61.43_{\pm1.1}$(27.8) | $57.35_{\pm1.1}$(32.6) | 25.5 |
| | DySAT | $78.52_{\pm0.4}$ | $70.64_{\pm0.1}$(10.0) | $63.35_{\pm0.0}$(19.3) | $56.36_{\pm0.0}$(28.2) | 19.2 | $66.21_{\pm0.9}$(15.7) | $56.28_{\pm0.9}$(28.3) | $53.45_{\pm1.1}$(31.9) | 25.3 |
| | SpoT-Mamba | $73.29_{\pm1.2}$ | $65.64_{\pm1.1}$(10.4) | $61.99_{\pm0.9}$(15.4) | $51.08_{\pm0.8}$(30.3) | 18.7 | $62.89_{\pm0.9}$(14.9) | $58.04_{\pm1.3}$(20.8) | $51.04_{\pm1.2}$(30.4) | 22.0 |
| | TGSL | $80.53_{\pm0.5}$ | $72.26_{\pm0.3}$(12.8) | $67.34_{\pm0.2}$(16.4) | $61.55_{\pm0.3}$(23.6) | 17.6 | $68.10_{\pm0.9}$(15.4) | $61.07_{\pm1.0}$(24.2) | $59.39_{\pm1.0}$(26.3) | 22.0 |
| | WinGNN | $90.12_{\pm0.4}$ | $80.16_{\pm0.4}$(11.1) | $72.50_{\pm0.3}$(19.6) | $63.21_{\pm0.4}$(29.9) | 20.2 | $81.26_{\pm0.9}$(9.8) | $67.33_{\pm1.1}$(25.3) | $61.25_{\pm1.0}$(32.0) | 22.4 |
| | DGIB-Cat | $94.89_{\pm0.2}$ | $84.98_{\pm0.1}$(10.4) | $76.78_{\pm0.1}$(19.1) | $67.69_{\pm0.1}$(28.7) | 19.4 | $80.16_{\pm0.4}$(15.5) | $68.71_{\pm1.5}$(27.6) | $64.38_{\pm0.6}$(32.2) | 25.1 |
| | DG-Mamba | $96.67_{\pm0.3}$ | $86.62_{\pm0.1}$(10.4) | $85.58_{\pm0.1}$(11.5) | $67.12_{\pm0.5}$(30.6) | $\underline{17.5}$ | $85.53_{\pm0.6}$(11.5) | $75.62_{\pm0.2}$(21.8) | $65.65_{\pm0.5}$(32.1) | $\underline{21.8}$ |
| | DeR-Mamba | $\mathbf{97.17}_{\pm0.2}$ | $\mathbf{88.75}_{\pm0.4}$(8.7) | $\mathbf{88.10}_{\pm0.5}$(9.3) | $\mathbf{79.30}_{\pm0.6}$(18.4) | **12.1** | $\mathbf{88.46}_{\pm0.7}$(9.0) | $\mathbf{87.56}_{\pm0.5}$(9.9) | $\mathbf{79.02}_{\pm0.7}$(18.7) | **12.5** |

## 5.2. Robustness to Non-Targeted Adversarial Attacks

We evaluate the models on dynamic link prediction, examining robustness to non-targeted adversarial attacks, including structure and feature attacks. Table 1 presents the mean AUC (Area under the curve) scores averaged over five runs.

**Observations.** DeR-Mamba achieves the strongest performance under non-targeted adversarial attack settings. Static GNNs fail to capture spatiotemporal dynamics, while dynamic GNNs rely on sequential modeling, leading to accumulated structural noise. DGSL methods are sensitive to incomplete observations and uncertain structural evolution, resulting in degradation under perturbations. Robust DGNNs provide moderate noise resistance but remain limited by single-source denoising. DGIB marginally outperforms DeR-Mamba in few cases due to short-term modeling, whereas DG-Mamba mitigates noise via state-space modeling but is less robust to feature perturbations.

## 5.3. Robustness to Targeted Adversarial Attacks

Targeted attacks are more challenging due to deliberate structural exploitation. We assess the robustness of DeR-Mamba against the competitive baselines in Table 1 under targeted adversarial attacks, with results reported in Table 2.

**Observations.** Compared with competitive baselines, DeR-Mamba shows the smallest performance degradation under targeted adversarial attacks, demonstrating superior robustness. Static GNNs such as GAT, DySAT, and SpoT-Mamba are highly vulnerable to structural manipulation, resulting in pronounced AUC drops. Dynamic graph structure learning methods including TGSL provide limited stabilization through structural evolution modeling but still degrade severely under targeted attacks. Robust DGNNs such as WinGNN, DGIB, and DG-Mamba achieve moderate improvements, yet without multi-view noise decoupling their performance degrades sharply under stronger attacks.

*Table 3.* Ablation results on COLLAB under feature, evasion, and poisoning attacks. The best results are highlighted in **bold**.

| Model | Original | Feature Attack | | | Evasion Attack | | | | Poisoning Attack | | | |
|---|---|---|---|---|---|---|---|---|---|---|---|---|
| | | $\gamma=0.5$ | $\gamma=1.0$ | $\gamma=1.5$ | $n_p=1(\Delta\%\downarrow)$ | $n_p=2(\Delta\%\downarrow)$ | $n_p=3(\Delta\%\downarrow)$ | Avg. $\Delta\%\downarrow$ | $n_p=1(\Delta\%\downarrow)$ | $n_p=2(\Delta\%\downarrow)$ | $n_p=3(\Delta\%\downarrow)$ | Avg. $\Delta\%\downarrow$ |
| **DeR-Mamba** | $94.83_{\pm0.1}$ | $70.16_{\pm0.4}$ | $61.99_{\pm0.2}$ | $57.50_{\pm0.3}$ | $82.77_{\pm0.6}(12.7)$ | $81.13_{\pm0.7}(14.4)$ | $69.42_{\pm1.0}(26.8)$ | **18.0** | $82.53_{\pm0.3}(13.0)$ | $81.01_{\pm0.4}(14.6)$ | $68.38_{\pm0.4}(27.9)$ | **18.5** |
| DeR-Mamba (w/o MP–K²alman) | $91.79_{\pm0.1}$ | $69.17_{\pm0.4}$ | $60.98_{\pm0.2}$ | $56.51_{\pm0.1}$ | $72.05_{\pm0.5}(21.5)$ | $62.02_{\pm0.7}(32.4)$ | $53.75_{\pm0.8}(41.4)$ | 31.8 | $70.47_{\pm0.3}(23.2)$ | $60.15_{\pm0.5}(34.5)$ | $52.52_{\pm0.3}(42.8)$ | 33.5 |
| DeR-Mamba (w/o AFDM) | $94.06_{\pm0.1}$ | $69.35_{\pm0.2}$ | $61.59_{\pm0.1}$ | $57.34_{\pm0.2}$ | $78.41_{\pm0.7}(16.6)$ | $76.94_{\pm0.6}(18.2)$ | $65.02_{\pm1.2}(30.9)$ | 21.9 | $77.93_{\pm0.5}(17.1)$ | $76.44_{\pm0.6}(18.7)$ | $64.43_{\pm0.2}(31.5)$ | 22.4 |
| DeR-Mamba (w/o all) | $91.23_{\pm0.2}$ | $68.63_{\pm0.5}$ | $61.42_{\pm0.2}$ | $56.77_{\pm0.1}$ | $70.56_{\pm0.6}(22.7)$ | $61.22_{\pm0.6}(32.9)$ | $53.08_{\pm0.9}(41.8)$ | 32.5 | $68.88_{\pm0.6}(24.5)$ | $59.43_{\pm0.5}(34.9)$ | $51.92_{\pm0.4}(43.1)$ | 34.2 |

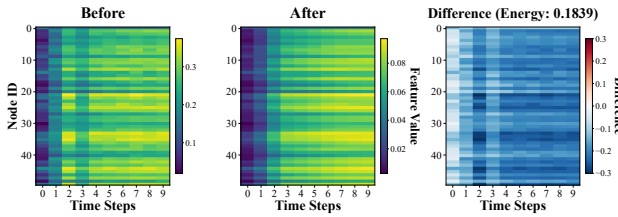

*Figure 3.* Ablation study results under original and structure attack.

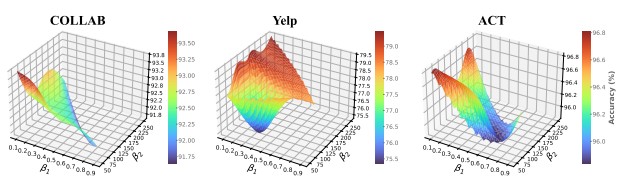

*Figure 4.* Spatiotemporal evolution of the Kalman gain.

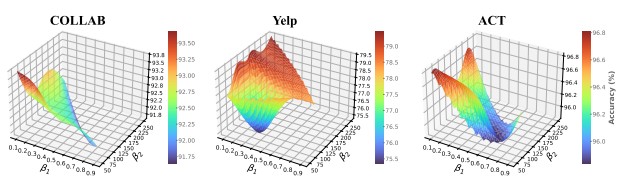

*Figure 5.* Feature maps before and after frequency decoupling.

*Figure 6.* Hyperparameter sensitivity under structure attack.

## 5.4. Ablation Studies

We perform ablation studies by removing MP–K²alman and AFDM from DeR-Mamba and evaluating the resulting variants under both non-targeted and targeted adversarial attacks. The results are summarized in Table 3 and Figure 3.

**Impact of MP–K²alman.** Removing MP–K²alman leads to consistent degradation across all settings. Specifically, drops of 3.04%, 2.73%, and 1.82% in original settings aggravate to 3.14%, 3.87%, and 1.90% under structure attacks. This confirms that multi-particle Kalman-style updates are critical for capturing structural uncertainty and maintaining robustness against incomplete or corrupted graph structures.

**Impact of AFDM.** Eliminating AFDM incurs notable drops of 4.36%, 4.19%, 4.40% (evasion) and 4.60%, 4.57%, 3.95% (poisoning) across strengths ($n_p = 1, 2, 3$). This confirms AFDM is vital for robustness against complex perturbations. By separating unstable high-frequency components, AFDM suppresses harmful temporal propagation and enhances robustness through frequency-domain discrimination.

**Joint effects.** The complete DeR-Mamba model maintains strong robustness across all attack scenarios. Notably, it limits the average performance degradation under evasion and poisoning attacks to only 18.0% and 18.5%, respectively. This advantage arises from the joint handling of structural uncertainty and noise perturbations, together with selective state-space modeling that mitigates temporal redundancy.

## 5.5. Efficiency Analysis

DeR-Mamba significantly outperforms baselines, reducing training latency by up to 49.8% and memory footprint by 56.5%. Further details are provided in Appendix A and F.1.

## 5.6. Interpretability and Visualization

**Visualization of Kalman gain evolution.** Figure 4 visualizes the converged Kalman gain of MP–K²alman. The gain exhibits a stable, stratified spatiotemporal structure, indicating differentiated node-level responsiveness and mitigation of spatial error amplification, reducing representation drift.

**Visualization of frequency decoupling.** Figure 5 visualizes feature maps before and after AFDM-based frequency decoupling and their differences. AFDM suppresses non-stationary fluctuations, yielding smoother temporal patterns, while the difference maps highlight frequency-selective adjustments across nodes and time. Further analysis under non-targeted and targeted attacks is provided in Appendix F.2.2

## 5.7. Hyperparameter Sensitivity Analysis

We analyze the sensitivity of the key hyperparameters $\beta_1$ and $\beta_2$ (Figure 6). $\beta_1$ controls residual suppression, while $\beta_2$ governs the shrinkage–KL trade-off. The model remains stable over a broad range, with a clear optimal region. Additional analysis is provided in Appendix F.5.

## 6. Conclusion

In this paper, we propose DeR-Mamba to enhance the robustness of DGNNs by decoupling spatial, spectral, and temporal dynamics. The framework incorporates MP–K²alman to correct structural estimation errors and AFDM to modulate non-stationary spectral perturbations. Combined with a redundancy-aware selective state-space mechanism, DeR-Mamba achieves superior robustness. Extensive evaluations demonstrate that our method consistently outperforms state-of-the-art baselines while maintaining competitive computational efficiency. In the future work, we will explore adapting DeR-Mamba to continuous-time dynamic graphs.

## Acknowledgements

This work was supported in part by the National Natural Science Foundation of China under Grants 62477016, 62377028 and 62077045, Guangdong Basic and Applied Basic Research Foundation under Grants 2024A1515011758 and 2024A1515140144, Key Research and Development Project of Guangdong Province, China (2025B0101120004), Science and Technology Planning Project of Guangzhou Development District (2023GH01), Master Mentor Plan of Jinan University (YDXS2501), the Fundamental Research Funds for the Central Universities (21625102), the Teaching Reform Research Projects of Jinan University (JG2026030) and the Fund of Key Laboratory of Education Blockchain and Intelligent Technology (Guangxi Normal University), Ministry of Education.

## Impact Statement

This paper presents work whose goal is to advance the field of Machine Learning. There are many potential societal consequences of our work, none of which we feel must be specifically highlighted here.

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

# A. Algorithms and Complexity Analysis

The complete training pipeline of DeR-Mamba is summarized in Algorithm 1, with the Dynamic Graph Redundancy-aware Scanning (DGRS) procedure detailed in Algorithm 2.

---

**Algorithm 1** Overall training pipeline of DeR-Mamba.

---

**Require:** A dynamic graph $\mathcal{G}^{1:T}$, accompanied by node features $\mathbf{X}^{1:T+1}$, and link occurrence labels $\mathbf{Y}^{1:T}$, the predictive model is denoted as $f_\theta = \psi \circ \varphi$.
**Parameter:** Number of training epochs $E$, network depth $L$, and hyperparameters $\beta_1$, $\beta_2$ and $M$.
**Output:** Optimized parameter $f_\theta^* = \psi \circ \varphi$; Predicted link occurrence label $\hat{\mathbf{Y}}^{T+1}$ at time step $T+1$.

1: Random initialization of the trainable parameters $\theta$;
2: Relative time encoding: $\mathbf{Z}^{1:T+1} \leftarrow \text{RTE}(\mathbf{X}^{1:T+1})$;
3: **for** $e \in \{1, \ldots, E\}$ **(epochs) do**
4:     **for** $l \in \{1, \ldots, L\}$ **(layers) do**
5:         *Spatial decoupling with MP–K$^2$alman:*
6:             Sample queries/values: $\{\mathbf{Q}_t^{(m,h)}, \mathbf{V}_t^{(m,h)}\} \leftarrow$ Eq. (2);
7:             Centralize queries/values: $\{\overline{\mathbf{Q}}_t^{(h)}, \overline{\mathbf{V}}_t^{(h)}\} \leftarrow$ Eqs. (3–4);
8:             Estimate var and cov: $\{\text{Var}_{q,t}^{(h)}, \text{Cov}_{qv,t}^{(h)}\} \leftarrow$ Eqs. (5–6);
9:             Kalman update of latent: $\mathbf{Z}_t^{(h)} \leftarrow$ Eqs. (7–8);
10:           Structure enhancement: $\mathbf{Z}_t^{(h)} \leftarrow$ Eqs. (9–10);
11:         *Spectral decoupling with AFDM:*
12:           Wavelet transform and frequency decomposition:
               $\{\mathbf{W}_{lo}, \mathbf{W}_{hi}, \mathbf{1}_{n,c}, \mathbf{h}_{n,c}\} \leftarrow$ Eqs. (11–13);
13:           Adaptive cross-frequency modulation: $\alpha_{n,c} \leftarrow$ Eqs. (15–16);
14:           Obtain decoupled representation: $\mathbf{Z}_{\text{Dec}} \leftarrow$ Eq. (17);
15:         *Temporal decoupling with SSM:*
16:           Average pooling: $\overline{\mathbf{Z}}_{\text{Dec}} \leftarrow \text{AvgPool}(\mathbf{Z}_{\text{Dec}})$;
17:           Selective state-space for long-range dependencies:
               $\mathbf{Z}_{\text{seq}} \leftarrow \text{DGRS}(\overline{\mathbf{Z}}_{\text{Dec}}, \hat{\mathbf{A}}_{\text{inter}}^{1:T})$ (see Algorithm 2);
18:           Robust temporal representation: $\hat{\mathbf{Z}} \leftarrow$ Eq. (26);
19:     **end for**
20:     Predict: $\hat{\mathbf{Y}}^{T+1} = \psi(\hat{\mathbf{Z}}^{T+1(L)})$;
21:     $\mathcal{L}_{\text{LP}} \leftarrow \text{CE}(\mathbf{Y}^{T+1}, \hat{\mathbf{Y}}^{T+1})$, $\mathcal{L}_{\text{mamba}}(\mathbf{Z}_{\text{Seq}}, \overline{\mathbf{Z}}_{\text{Dec}}) \leftarrow$ Eqs. (27–28);
22:     Evaluate the joint loss function: $\mathcal{L} \leftarrow \mathcal{L}_{\text{LP}} + \beta_1 \cdot \mathcal{L}_{\text{mamba}}$;
23:     Optimize $\theta$ by minimizing $\mathcal{L}$ through backpropagation.
24: **end for**

---

**Computational complexity analysis.** To present a concise and clear time complexity analysis of DeR-Mamba, we denote $|\mathcal{V}|$ as the average number of nodes and $|\mathcal{E}|$ as the average number of edges across all time steps in the dynamic graph, with $T$ representing the number of time steps. Let $d$ denote the input feature dimension of the nodes, $d_\phi$ the projection dimension obtained via Random Feature Projection, $C$ the number of hidden channels learned in the AFDM, and $D$ the feature dimension of the latent state representation. We detail the computational complexity of each component of DeR-Mamba during the training process as follows:

- Relative Time Encoding (RTE) layer: $\mathcal{O}(T|\mathcal{V}|d^2)$

- Linear input feature projection layer: $\mathcal{O}(T|\mathcal{V}|dd_\phi)$

- Multi-particle Kernelized Kalman Observation Field (MP–K$^2$alman): $\mathcal{O}(MT(|\mathcal{V}| + |\mathcal{E}|)d_\phi)$

- Adversarial-aware Frequency Decoupling Module (AFDM): $\mathcal{O}(TC \log C)$

- Selective State-space for Long-range Dependencies: $\mathcal{O}(TD)$

---

**Algorithm 2** Dynamic Graph Redundancy-aware Scan (DGRS).

---

**Require:** Average-pooled representation $\overline{\mathbf{Z}}_{\mathrm{Dec}} \in \mathbb{R}^{B \times T \times N}$;
Inter-graph structural matrices $\hat{\mathbf{A}}_{\mathrm{inter}}^{1:T} \in \mathbb{R}^{B \times T \times N \times N}$.
**Notation:** Batch size $B$, time steps $T$, number of latent states $N$, hidden dimension $D$, weight matrix $\mathbf{W}$.
**Output:** Redundancy-suppressed temporal representation capturing long-range dependencies $\mathbf{Z}_{\mathrm{Seq}} \in \mathbb{R}^{B \times T \times N}$.

1: Initialize transition matrix $\mathcal{A} \in \mathbb{R}^{N \times D}$;
2: Produce projection matrices $\mathcal{B}, \mathcal{C} \in \mathbb{R}^{B \times T \times D} \leftarrow \mathrm{Linear}_D(\overline{\mathbf{Z}}_{\mathrm{Dec}})$;
3: Compute step size $\Delta \in \mathbb{R}^{B \times T \times N} \leftarrow \mathrm{softplus}(\mathrm{Linear}_N(\overline{\mathbf{Z}}_{\mathrm{Dec}}))$;
4: Refine $\Delta$ via redundancy-aware structural alignment:
$$\overline{\Delta} \leftarrow \mathrm{unsqueeze}_N(\Delta) \cdot \mathbf{W}\hat{\mathbf{A}}_{\mathrm{inter}}^{1:T};$$
5: Discretize transition $\overline{\mathbf{A}} \leftarrow \exp(\overline{\Delta}\mathbf{A})$ using ZOH–Taylor ;
6: Update $\overline{\mathbf{B}} \in \mathbb{R}^{B \times T \times N \times D} \leftarrow (\Delta\mathbf{A})^{-1}(\exp(\Delta\mathbf{A}) - \mathbf{I})\Delta\mathbf{B}$;
7: Initialize latent state $\mathbf{H} \in \mathbb{R}^{B \times T \times N \times D}$ to zero;
8: Initialize output $\mathbf{Z}_{\mathrm{Seq}} \in \mathbb{R}^{B \times T \times N}$ as an empty tensor;
9: **for** $t = 1$ to $T - 1$ **do**
10: Update latent state: $\mathbf{H}_t \leftarrow \overline{\mathbf{A}}_t\mathbf{H}_{t-1} + \overline{\mathbf{B}}(\overline{\mathbf{Z}}_{\mathrm{Dec}})_t$;
11: Project to output: $(\mathbf{Z}_{\mathrm{Seq}})_t \leftarrow \mathbf{C}_t\mathbf{H}_t$;
12: **end for**
13: **return** $\mathbf{Z}_{\mathrm{Seq}} \in \mathbb{R}^{B \times T \times N}$ // redundancy-suppressed sequence

---

- Link prediction module (Multilayer perceptron), feature fusion, and nonlinear activations: Their computational costs scale only with the latent feature dimensions. Since these dimensions are relatively small, they can be treated as constants in the overall complexity analysis and are therefore negligible.

To sum up, the total complexity of DeR-Mamba is formulated as:

$$\mathcal{O}(T|\mathcal{V}|d^2) + \mathcal{O}(T|\mathcal{V}|dd_\phi) + \mathcal{O}(MT(|\mathcal{V}| + |\mathcal{E}|)d_\phi) + \mathcal{O}(TC\log C) + \mathcal{O}(TD). \tag{29}$$

Since the feature dimensions $d, d_\phi, C, D$, and the number of sampled particles $M$ are typically much smaller compared with the average numbers of nodes $|\mathcal{V}|$ and edges $|\mathcal{E}|$, their contributions to the overall cost are negligible. Consequently, the overall computational complexity is primarily governed by the graph scale and can be approximated as follows:

$$\begin{aligned}
&\mathcal{O}(T|\mathcal{V}|) + \mathcal{O}(T(|\mathcal{V}| + |\mathcal{E}|)) + \mathcal{O}(T) \\
&= \mathcal{O}(T(|\mathcal{V}| + |\mathcal{E}|)).
\end{aligned} \tag{30}$$

In summary, the overall computational complexity of DeR-Mamba maintains linearity. Its growth is primarily constrained by the number of nodes, edges, and the scale of time steps, rendering DeR-Mamba highly efficient when processing large-scale dynamic graphs. Unlike many DGNNs that rely on global convolution or attention mechanisms, which suffer from low computational efficiency on sparse graph structures, DeR-Mamba retains linear complexity even in sparse scenarios. Consequently, it demonstrates significant advantages on large-scale real-world dynamic graphs. Furthermore, we detail the hardware configurations required based on our experimental experience, including memory requirements and corresponding computing power, in Appendix J.2. Detailed empirical evaluations, including node and sequence scalability tests that further validate this linear efficiency, are provided in Appendix F.1.

## B. Proof

### B.1. Proof 1

**Theorem 1 (Information-Theoretic Stability & Optimality).** Under the standard assumption of Linear Gaussian dynamics, the fused representation $\mathbf{z}_t$ exhibits both stability and optimality. Its information capacity (quantified by the precision matrix) is lower-bounded by the confidence-weighted average of all particle trajectories (*Stability*) and asymptotically converges to the capacity of the optimal particle trajectory (*Optimality*):

$$I(\mathcal{G}_t; \mathbf{z}_t) \geq \sum_{m=1}^{M} \rho^{(m)} I(\mathcal{G}_t; \mathbf{x}_t^{(m)}) \xrightarrow{\rho \to \rho^*} \max_m I(\mathcal{G}_t; \mathbf{x}_t^{(m)}), \tag{31}$$

where $I(\cdot; \cdot)$ denotes Mutual Information, $\mathcal{G}_t$ represents the ground-truth structural semantics, and $\rho^{(m)}$ is the normalized confidence weight assigned to the $m$-th particle. Note that for rigorous probabilistic derivation, we denote the latent variables as vectors (e.g., $\mathbf{z}_t$, $\mathbf{x}_t$) corresponding to the feature matrices ($\mathbf{Z}_t$, $\mathbf{X}_t$) in the main text.

**Proof sketch.** Here, we present the full derivations for stability and optimality, verifying that MP–K$^2$alman provides a robust lower bound while targeting the optimal structural semantics.

**(i) Stability via convexity of the information manifold.** To rigorously establish the stability lower bound, we analyze the spectral properties of the fused representation under the standard Linear Gaussian assumption. Let the posterior distribution of the $m$-th particle be modeled as a multivariate Gaussian $\mathbf{x}_t^{(m)} \sim \mathcal{N}(\boldsymbol{\mu}_m, \boldsymbol{\Sigma}_m)$. The differential entropy of $\mathbf{x}_t^{(m)}$ is fully determined by its precision matrix $\mathbf{P}_m = \boldsymbol{\Sigma}_m^{-1}$:

$$\mathcal{H}(\mathcal{G}_t | \mathbf{x}_t^{(m)}) = \frac{1}{2} \log |2\pi e \boldsymbol{\Sigma}_m| = c - \frac{1}{2} \log \det(\mathbf{P}_m), \tag{32}$$

where $c = \frac{d}{2} \log(2\pi e)$ is a dimension-dependent constant ($d$ is the state dimension). We formulate the Kalman-style fusion in MP-K$^2$alman as an instantiation of the Generalized Product of Experts (gPoE) framework (Cao & Fleet, 2014). Specifically, the fused posterior is defined as the normalized product of expert densities raised to their confidence powers $\rho^{(m)}$ (where $\sum \rho^{(m)} = 1$). Consequently, the effective precision matrix of the fused state $\mathbf{z}_t$, denoted as $\mathbf{P}_z$, is exactly the convex combination of individual precisions:

$$\mathbf{P}_z = \sum_{m=1}^{M} \rho^{(m)} \mathbf{P}_m. \tag{33}$$

Leveraging the Minkowski Determinant Theorem (Horn & Johnson, 2012), we utilize the property that the log-determinant function $f(\mathbf{P}) = \log \det(\mathbf{P})$ is strictly concave on the cone of positive definite matrices. Applying Jensen's Inequality to the convex combination in Eq. (33) yields:

$$\log \det(\mathbf{P}_z) = \log \det \left( \sum_{m=1}^{M} \rho^{(m)} \mathbf{P}_m \right) \geq \sum_{m=1}^{M} \rho^{(m)} \log \det(\mathbf{P}_m). \tag{34}$$

Substituting this into the entropy definition (note that the negative sign reverses the inequality):

$$\mathcal{H}(\mathcal{G}_t | \mathbf{z}_t) = c - \frac{1}{2} \log \det(\mathbf{P}_z) \leq \sum_{m=1}^{M} \rho^{(m)} \left( c - \frac{1}{2} \log \det(\mathbf{P}_m) \right) = \sum_{m=1}^{M} \rho^{(m)} \mathcal{H}(\mathcal{G}_t | \mathbf{x}_t^{(m)}). \tag{35}$$

Finally, since the entropy of the ground-truth structure $\mathcal{H}(\mathcal{G}_t)$ is invariant to the estimation process, the Mutual Information lower bound is derived as:

$$
\begin{aligned}
I(\mathcal{G}_t; \mathbf{z}_t) &= \mathcal{H}(\mathcal{G}_t) - \mathcal{H}(\mathcal{G}_t | \mathbf{z}_t) \\
&\geq \mathcal{H}(\mathcal{G}_t) - \sum_{m=1}^{M} \rho^{(m)} \mathcal{H}(\mathcal{G}_t | \mathbf{x}_t^{(m)}) \\
&= \sum_{m=1}^{M} \rho^{(m)} \left( \mathcal{H}(\mathcal{G}_t) - \mathcal{H}(\mathcal{G}_t | \mathbf{x}_t^{(m)}) \right) \\
&= \sum_{m=1}^{M} \rho^{(m)} I(\mathcal{G}_t; \mathbf{x}_t^{(m)}).
\end{aligned}
\tag{36}
$$

This confirms that the fused representation $\mathbf{z}_t$ maintains a stability floor strictly bounded by the ensemble's weighted average information capacity.

**(ii) Optimality via variational bound (the aligned max bound).** While Eq. (36) guarantees a stability floor based on the ensemble average, it does not explicitly characterize the convergence behavior towards the best particle. To strictly prove the asymptotic optimality, we now analyze the variational lower bound behavior under sharpening attention. Let $q(\mathcal{G}_t | \mathbf{z}_t)$ denote the variational posterior approximation modeled as a mixture of particles. Exploiting the Log-Sum-Exp inequality $\log(\sum a_i) \geq \max_i(\log a_i)$ and Jensen's Inequality $\mathbb{E}[\max(\cdot)] \geq \max(\mathbb{E}[\cdot])$, we derive:

$$
\begin{aligned}
I(\mathcal{G}_t; \mathbf{z}_t) &\geq \mathbb{E}_{p(\mathcal{G}_t, \mathbf{z}_t)} \left[ \log q(\mathcal{G}_t | \mathbf{z}_t) \right] + \mathcal{H}(\mathcal{G}_t) \\
&= \mathbb{E} \left[ \log \left( \sum_{m=1}^{M} \rho^{(m)} q(\mathcal{G}_t | \mathbf{x}_t^{(m)}) \right) \right] + \mathcal{H}(\mathcal{G}_t) \\
&\geq \mathbb{E} \left[ \max_m \left( \log \rho^{(m)} + \log q(\mathcal{G}_t | \mathbf{x}_t^{(m)}) \right) \right] + \mathcal{H}(\mathcal{G}_t) \\
&\geq \max_m \left( \mathbb{E}_{p(\mathcal{G}_t, \mathbf{x}_t^{(m)})} \left[ \log q(\mathcal{G}_t | \mathbf{x}_t^{(m)}) \right] + \log \rho^{(m)} \right) + \mathcal{H}(\mathcal{G}_t) \\
&= \max_m \left( I(\mathcal{G}_t; \mathbf{x}_t^{(m)}) - \underbrace{(-\log \rho^{(m)})}_{\Omega^{(m)}} \right), \quad \text{where } \Omega^{(m)} \geq 0.
\end{aligned}
\tag{37}
$$

Eq. (37) reveals that the system capacity is bounded by the best particle's information minus a non-negative term $\Omega^{(m)}$, which we term the attention penalty. This penalty reflects the information cost of uncertain particle selection. Let $k^* = \arg\max_m I(\mathcal{G}_t; \mathbf{x}_t^{(m)})$ denote the index of the optimal particle. As the Kalman gain sharpens the attention weights (i.e., confidence $\rho$ concentrates on the reliable particle), we have $\rho \to \mathbf{e}_{k^*}$. Consequently, the penalty for the optimal particle vanishes ($\Omega^{(k^*)} = -\log(1) = 0$), while it approaches infinity for others. Thus, the bound relies solely on the optimal term:

$$
\max_m \left( I(\mathcal{G}_t; \mathbf{x}_t^{(m)}) - \Omega^{(m)} \right) \xrightarrow{\rho \to \mathbf{e}_{k^*}} I(\mathcal{G}_t; \mathbf{x}_t^{(k^*)}) \equiv \max_m I(\mathcal{G}_t; \mathbf{x}_t^{(m)}).
\tag{38}
$$

This completes the proof of the optimality convergence. $\qquad\square$

## B.2. Proof 2

**Proposition 1 (Spectral Non-Expansiveness).** *The AFDM modulation operator $\mathcal{M} : \mathbf{l}_{n,c} \mapsto \mathbf{y}_{n,c}$ is structurally non-expansive with respect to the $L_2$-norm, satisfying $\|\mathbf{y}_{n,c}\|_2 \leq \|\mathbf{l}_{n,c}\|_2$ for any input state.*

*Proof.* By construction, the adversarial weights lie in the probability simplex $\boldsymbol{\alpha}_{n,c} \in \mathcal{U}^{T-1}$, implying $\|\boldsymbol{\alpha}_{n,c}\|_\infty \leq 1$. Considering the element-wise operation $\mathbf{y}_{n,c}[t] = \mathbf{l}_{n,c}[t] \cdot \boldsymbol{\alpha}_{n,c}[t]$, we have:

$$
\begin{aligned}
\|\mathbf{y}_{n,c}\|_2^2 &= \sum_{t=1}^{T} (\mathbf{l}_{n,c}[t] \cdot \boldsymbol{\alpha}_{n,c}[t])^2 \\
&\leq \left( \max_t |\boldsymbol{\alpha}_{n,c}[t]| \right)^2 \sum_{t=1}^{T} (\mathbf{l}_{n,c}[t])^2 \\
&\leq \|\mathbf{l}_{n,c}\|_2^2.
\end{aligned}
\tag{39}
$$

This confirms that the modulation strictly bounds the spectral energy, preventing the system from entering unstable high-energy states due to cross-frequency resonance. $\qquad\square$

**Proposition 2 (Differential Perturbation Stability).** *Let $\tilde{\mathbf{x}} = \mathbf{x} + \boldsymbol{\delta}$ be the perturbed input. The resulting perturbation in the modulated output $\|\Delta\mathbf{y}_{n,c}\|_2$ is bounded by a decoupled error term:*

$$
\|\Delta\mathbf{y}_{n,c}\|_2 \leq \underbrace{\|\Delta\mathbf{l}_{n,c}\|_2}_{\text{Additive Noise}} + \underbrace{\frac{1}{\lambda} \|\Delta\mathbf{h}_{n,c}\|_2 \|\mathbf{l}_{n,c}\|_\infty}_{\text{Gated Multiplicative Noise}},
\tag{40}
$$

*where $\Delta\mathbf{l}_{n,c}$ and $\Delta\mathbf{h}_{n,c}$ denote the induced perturbations in the low-frequency and high-frequency components, and $\lambda$ is the regularization temperature.*

*Proof.* The adversarial weight is given by $\boldsymbol{\alpha} = G(\mathbf{H}/\lambda)$, where $G(\cdot)$ denotes the Gibbs distribution function (Eq. (16)). We apply the first-order perturbation expansion (Taylor expansion) to the modulated output $\mathbf{y}$: $\Delta\mathbf{y} \approx (\Delta\mathbf{l}) \odot \boldsymbol{\alpha} + \mathbf{l} \odot (\Delta\boldsymbol{\alpha})$. Applying the Triangle Inequality and Cauchy-Schwarz Inequality yields:

$$
\begin{aligned}
\|\Delta\mathbf{y}\|_2 &\leq \|\Delta\mathbf{l} \odot \boldsymbol{\alpha}\|_2 + \|\mathbf{l} \odot \Delta\boldsymbol{\alpha}\|_2 \\
&\leq \|\Delta\mathbf{l}\|_2 \|\boldsymbol{\alpha}\|_\infty + \|\mathbf{l}\|_\infty \|\Delta\boldsymbol{\alpha}\|_2.
\end{aligned}
\tag{41}
$$

Since the Gibbs distribution is equivalent to the Softmax function, its derivative bound, specifically the Lipschitz constant $\mathcal{L}_\sigma$, is scaled by the temperature factor $1/\lambda$. The perturbation in the weight $\boldsymbol{\alpha}$ is thus bounded by $\|\Delta\boldsymbol{\alpha}\|_2 \leq \frac{1}{\lambda}\|\Delta\mathbf{h}\|_2$. Substituting the non-expansiveness bound $\|\boldsymbol{\alpha}\|_\infty \leq 1$ (from Proposition 1), we obtain the upper bound (Eq. (40)). This result confirms that high-frequency noise ($\Delta\mathbf{h}$) is suppressed through two mechanisms: the linear damping term $\|\Delta\mathbf{l}\|_2$, and the multiplicative gating factor $\|\mathbf{l}\|_\infty$. This mechanism prevents $\Delta\mathbf{h}$ from being directly added to the output, as typically seen in additive accumulation models, ensuring stability even when the temperature factor $1/\lambda$ increases the local gradient. This robust, multiplicative decoupling preserves spectral stability in stationary regimes. □

**Theoretical Analysis: Multiplicative gating vs. Additive accumulation.** To further validate the effectiveness of AFDM, we contrast it with standard temporal propagation from a Lipschitz perspective. Let $\mathcal{F}$ be a first-order temporal propagation operator (e.g., convolution or message passing) with Lipschitz constant $\kappa = |\mathcal{F}|$. If an input at time $\tau$ contains a perturbation $\delta_\tau$, the error upper bound after $t$ steps without filtering is:

$$\|\Delta\mathbf{x}_t\| \leq \sum_{\tau \leq t} \kappa^{t-\tau}\|\delta_\tau\|. \tag{42}$$

This represents a standard additive error accumulation. In contrast, with AFDM, the perturbation is decomposed into spectral components and propagated via multiplicative gating. By leveraging the bounded Jacobian of softmax, there exists a constant $C > 0$ such that:

$$\|\Delta\mathbf{y}_t\| \leq \sum_{\tau \leq t} \kappa^{t-\tau}(\|\Delta\mathbf{l}_\tau\| + C\|\Delta\mathbf{h}_\tau\|\|\mathbf{l}_\tau\|). \tag{43}$$

This restricts the cumulative impact of non-stationary high-frequency perturbations to the multiplicative gating term, avoiding the cascading amplification typical of additive injection.

## C. Observation Model and Bayesian Inference

We formalize the observation of dynamic graphs as an encoding process subject to multi-domain entanglement. The observed graph state at time $t$, denoted as $\tilde{\mathcal{G}}_t$, is defined as:

$$\tilde{\mathcal{G}}_t = \mathcal{G}_t + \varepsilon_t, \quad \varepsilon_t \sim \mathcal{D}_{\text{perturb}}, \tag{44}$$

where $\mathcal{G}_t$ is the ground-truth structure, and $\varepsilon_t$ represents the perturbation component (e.g., adversarial edges or observation gaps) following distribution $\mathcal{D}_{\text{perturb}}$. Our goal is to infer the latent true structure $\mathcal{G}_t$ given noisy observations $\tilde{\mathcal{G}}_{1:t}$. According to Bayes' theorem (Bayes, 1958), the posterior distribution of the true structure is formulated as:

$$P(\mathcal{G}_t \mid \tilde{\mathcal{G}}_{1:t}) \propto P(\tilde{\mathcal{G}}_{1:t} \mid \mathcal{G}_t)P(\mathcal{G}_t), \tag{45}$$

where $P(\tilde{\mathcal{G}}_{1:t} \mid \mathcal{G}_t)$ represents the likelihood of observing the current sequence given the true structure, and $P(\mathcal{G}_t)$ reflects the prior assumption. Consequently, we employ Maximum A Posteriori (Bishop & Nasrabadi, 2006) (MAP) estimation to identify the most probable structural state:

$$\begin{aligned}
\hat{\mathcal{G}}_t &= \underset{\mathcal{G}_t}{\arg\max}\, P(\mathcal{G}_t \mid \tilde{\mathcal{G}}_{1:t}) \\
&= \underset{\mathcal{G}_t}{\arg\max}\, P(\tilde{\mathcal{G}}_{1:t} \mid \mathcal{G}_t)P(\mathcal{G}_t).
\end{aligned} \tag{46}$$

Direct optimization of Eq. (46) is intractable due to the high-dimensional, sparse combinatorial space of dynamic graphs. To address this, we introduce a multi-particle kernel subspace approximation.

## D. Datasets

We evaluate DeR-Mamba on the downstream task of dynamic link prediction across three real-world dynamic graph datasets. These datasets span diverse temporal ranges: 16 years, 24 months, and 30 days, thus forming a multi-scale real-world evaluation setting. Among them, COLLAB (Tang et al., 2012) presents the greatest challenge due to its long temporal span and coarse temporal granularity, which amplify issues such as structural incompleteness, spectral perturbations, and

accumulated historical redundancy. In addition, the Yelp (Sankar et al., 2020) and ACT (Kumar et al., 2019) datasets exhibit significant variations in edge attributes, further testing the model's robustness and generalization under perturbations. Detailed statistics of the three datasets are summarized in Table 4.

- **COLLAB** (Tang et al., 2012): This dataset is derived from an academic collaboration network, covering publications between 1990 and 2006. It contains 16 temporal graph snapshots that record the evolution of citation and collaboration relationships among researchers. Here, nodes denote authors and edges represent co-authorships. Each co-authorship is annotated with the research domain of the joint publication, such as Data Mining, Database, Medical Informatics, Theory, and Visualization. Following common practice, we extract 32-dimensional node representations from paper abstracts using the word2vec model (Mikolov et al., 2013).

- **Yelp** (Sankar et al., 2020): This dataset comes from a well-known crowd-sourced review platform, capturing interactions between customers and businesses from January 2019 to December 2020. It is organized into 24 temporal snapshots. The bipartite network contains customers and businesses as nodes, while edges denote review events. Each edge is further categorized into one of five business types: Pizza, American (New) Food, Coffee, Tea, Sushi Bars, and Fast Food. We leverage word2vec on the textual reviews to construct 32-dimensional node embeddings.

- **ACT** (Kumar et al., 2019): This dataset records learning activities in a large-scale MOOC platform over 30 consecutive days. The dynamic graphs capture interactions where nodes correspond to students or action targets, and edges represent student activities. To characterize these actions, we first cluster the raw features into five categories using K-Means, and then expand the initial 4-dimensional representations to 32-dimensional vectors via a linear transformation.

*Table 4.* Statistics of the dynamic graph datasets.

| Dataset | #Node | #Link | #Link Type | Length (Splits) | Temporal Granularity |
|---------|-------|-------|------------|-----------------|----------------------|
| COLLAB | 23,035 | 151,790 | 5 | 16 (10/1/5) | year |
| Yelp | 13,095 | 65,375 | 5 | 24 (15/1/8) | month |
| ACT | 20,408 | 202,339 | 5 | 30 (20/2/8) | day |

## E. Baselines Descriptions

In this section, we provide an overview of the baseline models used in our experiments. All hyperparameters are initialized according to the recommended settings reported in their original papers and are further fine-tuned where necessary to ensure fairness and reliability of comparison.

■ **Static GNNs.** To adapt static GNNs to dynamic settings, we extend them by stacking GNN layers with LSTM modules (Hochreiter & Schmidhuber, 1997), enabling temporal modeling:

- **GAE** (Kipf & Welling, 2016). A pioneering framework that leverages Graph Convolutional Networks (GCNs) to learn node embeddings and reconstruct adjacency relations. Although originally designed for static graphs, it has been widely used for graph representation learning tasks, including node clustering and link prediction.
- **VGAE** (Kipf & Welling, 2016). Extends GAE by embedding probabilistic reasoning into the latent space. Through a variational formulation, it can model uncertainty in node representations, making it more suitable for noisy or incomplete graph scenarios.
- **GAT** (Velickovic et al., 2018). Moves beyond uniform neighborhood aggregation by assigning data-driven attention weights. This mechanism enhances the model's expressiveness on graphs with irregular connectivity and has since become a widely used baseline across diverse graph learning tasks, including classification and link prediction.

■ **Dynamic GNNs (DGNNs)**

- **GCRN** (Seo et al., 2018). Integrates Graph Convolutional Networks with recurrent neural modules to capture both structural and temporal dependencies in dynamic graphs. By modeling spatial and sequential patterns jointly, GCRN provides a natural foundation for time-series forecasting and dynamic graph prediction.

- **EvolveGCN** (Pareja et al., 2020). Introduces an adaptive mechanism that evolves GCN parameters over time, replacing static weights with recurrently updated ones. This design enables the model to continually adjust to graph evolution, improving robustness and predictive accuracy in dynamic settings.
- **DySAT** (Sankar et al., 2020). Employs hierarchical self-attention across temporal and structural dimensions, allowing node embeddings to reflect long-range dependencies. DySAT is particularly effective in scenarios where interaction patterns shift frequently, making it a strong baseline for dynamic graph representation learning.
- **SpoT-Mamba** (Choi et al., 2024). Builds upon the state-space model Mamba (Gu & Dao, 2024) to capture long-horizon dependencies in spatio-temporal graphs. By combining structured walk-based encodings with temporal scanning, SpoT-Mamba generates expressive node embeddings tailored for forecasting tasks with complex long-range dynamics.

■ **Dynamic Graph Structure Learning (DGSL)**

- **RDGSL** (Zhang et al., 2023b). RDGSL introduces a dynamic filtering mechanism coupled with a temporal embedding learner to counteract noise in evolving graphs. Instead of treating all interactions equally, it emphasizes denoising and suppressing disruptive signals, thereby enhancing stability and yielding significant gains in classification performance across dynamic environments.
- **TGSL** (Zhang et al., 2023a). TGSL extends the framework of Temporal Graph Networks (TGNs) (Rossi et al., 2020) by explicitly modeling time-dependent edges. This design aims to mitigate the issues of noisy or missing connections in temporal graphs. By refining graph structures in an end-to-end manner, TGSL delivers stronger representations and achieves notable improvements on temporal prediction tasks.

■ **Robust DGNNs**

- **RGCN** (Zhu et al., 2019). RGCN strengthens the resilience of GCNs against adversarial perturbations by modeling node features with Gaussian distributions and employing a variance-aware attention mechanism. This probabilistic design enables the model to mitigate adversarial effects and significantly boost classification accuracy on benchmark datasets. In our dynamic setting, RGCN is extended by coupling GNN layers with vanilla LSTMs to capture temporal dependencies.
- **WinGNN** (Zhu et al., 2023). WinGNN introduces a meta-learning paradigm combined with random gradient aggregation to efficiently capture temporal dynamics in GNNs. By replacing explicit temporal encoders with a randomized sliding-window scheme for gradient updates, it reduces computational overhead while enhancing robustness. This design demonstrates strong adaptability and superior performance across diverse dynamic graph benchmarks.
- **DGIB** (Yuan et al., 2024). DGIB leverages the Information Bottleneck principle to improve the robustness of dynamic GNNs. By enforcing minimal and sufficient representations that remain stable under perturbations, DGIB enhances resistance to adversarial attacks. Its design facilitates more efficient information propagation in evolving graph environments and shows notable improvements on link prediction tasks.
- **DG-Mamba** (Yuan et al., 2025). DG-Mamba introduces a kernelized message-passing operator with linear complexity and a selective state-space module to capture long-range dependencies. Regularized by a principle of relevant information, it achieves robust dynamic graph modeling under noise and perturbations.

## F. Additional Experiment Results

### F.1. Additional Analysis: Scaling Efficiency Analysis

To comprehensively evaluate the efficiency of DeR-Mamba, we generate synthetic datasets by varying both node scale and sequence length. Our evaluation focuses on two primary dimensions: node scalability and sequence length scaling. We benchmark DeR-Mamba against two state-of-the-art and highly competitive baselines, DGIB and DG-Mamba.

**Node scalability.** We plot the training time of different methods across varying node scales on the COLLAB, Yelp, and ACT datasets, as shown in Figure 7. DeR-Mamba consistently achieves the lowest training time across all datasets, demonstrating superior scalability compared to the baselines. Specifically, DeR-Mamba reduces training overhead by up to 49.8% and 8.9% compared to DGIB and DG-Mamba, respectively. Furthermore, we evaluated the GPU memory footprint across different node scales on the Yelp dataset, as depicted in Figure 8. Compared to DGIB, DeR-Mamba reduces GPU memory consumption by approximately 56.5%. It maintains a memory footprint comparable to DG-Mamba, which utilizes

linear-complexity message-passing operators. DeR-Mamba exhibits linear growth in both training latency and memory usage, highlighting its exceptional efficiency for large-scale dynamic graph learning.

**Sequence length scaling.** We report the training latency of different methods on the Yelp dataset with sequence lengths scaled from $1\times$ to $8\times$, as shown in Figure 9, where blue crosses ($\times$) indicate Out-of-Memory (OOM) errors. DeR-Mamba maintains strict linear scalability with respect to sequence length and consistently outperforms all baselines in training speed. While DGIB encounters OOM errors at merely $2\times$ sequence length, DeR-Mamba remains operational within GPU memory constraints across the entire $1\times$ to $8\times$ range. This demonstrates its significant efficiency advantage in handling long-sequence graphs.

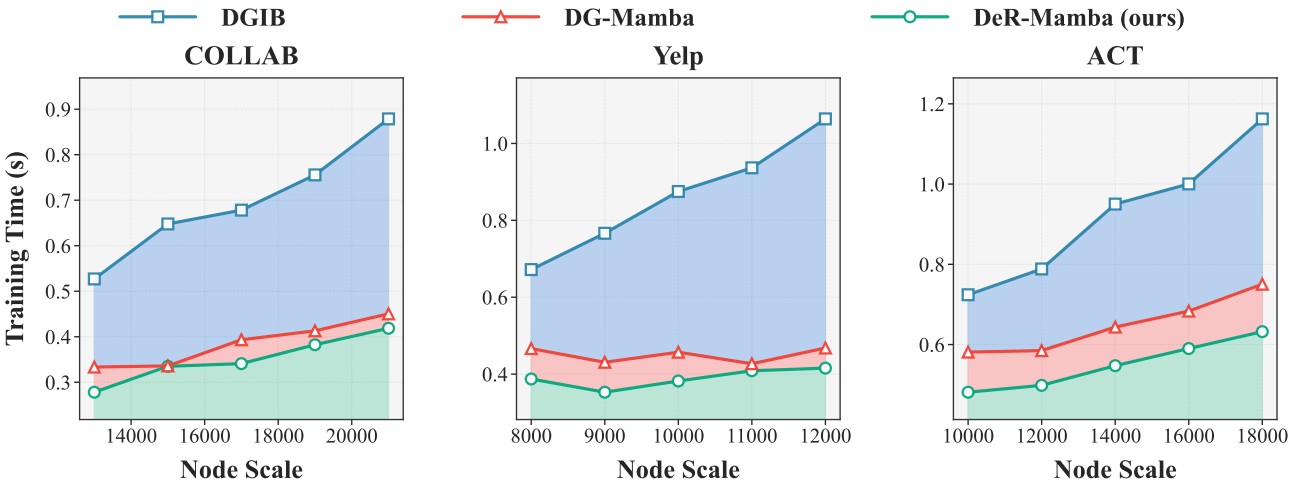

*Figure 7.* Training time comparison by increasing node scale.

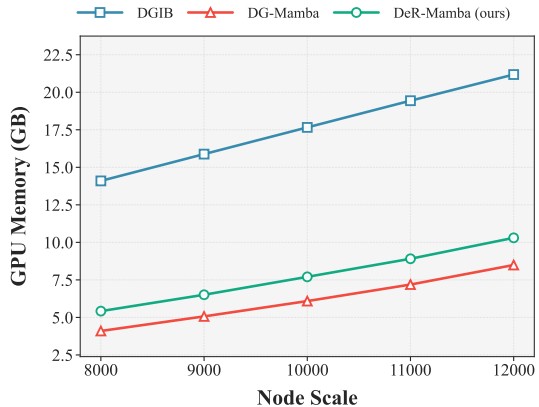

*Figure 8.* GPU memory comparison by increasing node scale on Yelp.

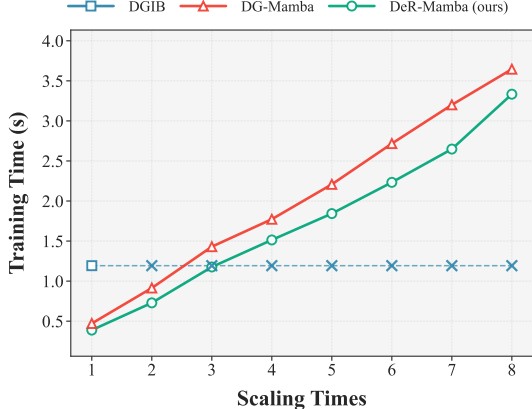

*Figure 9.* Training time across sequence scaling on Yelp

## F.2. Additional Results: Interpretability and Visualization

To provide intuitive insights into the working mechanisms of DeR-Mamba and the principles underlying its robustness, this section presents an in-depth analysis from the perspectives of interpretability and visualization. We focus on two core components:

- We analyze the evolution of the Kalman gain to illustrate how MP–K$^2$alman dynamically balances observation and prediction under structurally incomplete graph scenarios, thereby mitigating representation drift induced by missing or unreliable structural observations.

- We visualize AFDM in the feature domain to investigate how frequency-domain decoupling suppresses noise-induced spectral perturbations arising from internal and external disturbances, thereby alleviating signal distortion diffusion across snapshots.

### F.2.1. Spatiotemporal Profiling of MP–K$^2$alman Gain Dynamics

To further validate how MP-K$^2$alman models structural uncertainty during graph evolution, we provide an extended visualization of the evolution of the Kalman gain during training. We first present the spatiotemporal evolution heatmaps on the COLLAB, Yelp, and ACT datasets in Figures 10, 11, and 12, contrasting the early training stages with the converged state. Additionally, we further showcase the empirical distributions and detailed temporal dynamics of the gain values upon convergence, as depicted in Figures 13, 14, and 15. The Kalman gain reflects the trade-off between "prediction" and "observation," and its dynamics reveal the model's capability to filter unreliable structural observations. Thus, the gain evolution provides insight into how MP–K$^2$alman mitigates instability caused by incomplete or corrupted graph structures and enhances robustness.

**Spatiotemporal heatmap analysis.** Early in the training phase, as depicted in Figures 10, 11, and 12, the Kalman gain values are relatively small and uniform, indicating that the model initially relies more on predictive priors to avoid being misled by unstable or noisy observations. As training converges, the heatmaps exhibit a stable, stratified spatiotemporal structure with selective amplification on reliable nodes and snapshots. This differentiation confirms the capability of MP–K$^2$alman to effectively distinguish meaningful evolutionary pathways from spurious message flows induced by structural incompleteness that amplify spatial errors.

**Empirical distribution insights.** The histograms in Figures 13(a)–15(a) reveal that the converged Kalman gains exhibit a distinct heavy-tailed, right-skewed distribution. Specifically, the consistently low mean values—$3.09 \times 10^{-3}$ for COLLAB, $2.49 \times 10^{-2}$ for Yelp, and $1.38 \times 10^{-2}$ for ACT—indicate a highly selective filtering regime. Rather than uniformly updating all nodes, the framework predominantly relies on predictive priors to maintain global stability. Substantial corrections (high gains) are strictly reserved for a sparse subset of nodes undergoing significant structural evolution. This mechanism effectively decouples meaningful dynamics from spurious noise, preventing the cascading diffusion of spatial errors while preserving representational robustness. Complementing the distribution analysis, Figures 13(b)–15(b) illustrate the temporal evolution of the mean Kalman gain. The trajectories exhibit distinct non-stationary fluctuations rather than a static baseline, indicating a time-aware filtering mechanism. Specifically, the peaks and valleys in the gain curves reflect the model's dynamic modulation of information fusion: increasing reliance on observations during moments of abrupt structural change (peaks) while suppressing observational noise via predictive priors during stable or unreliable intervals (valleys). This temporal adaptivity ensures that the system robustly tracks the evolving graph topology.

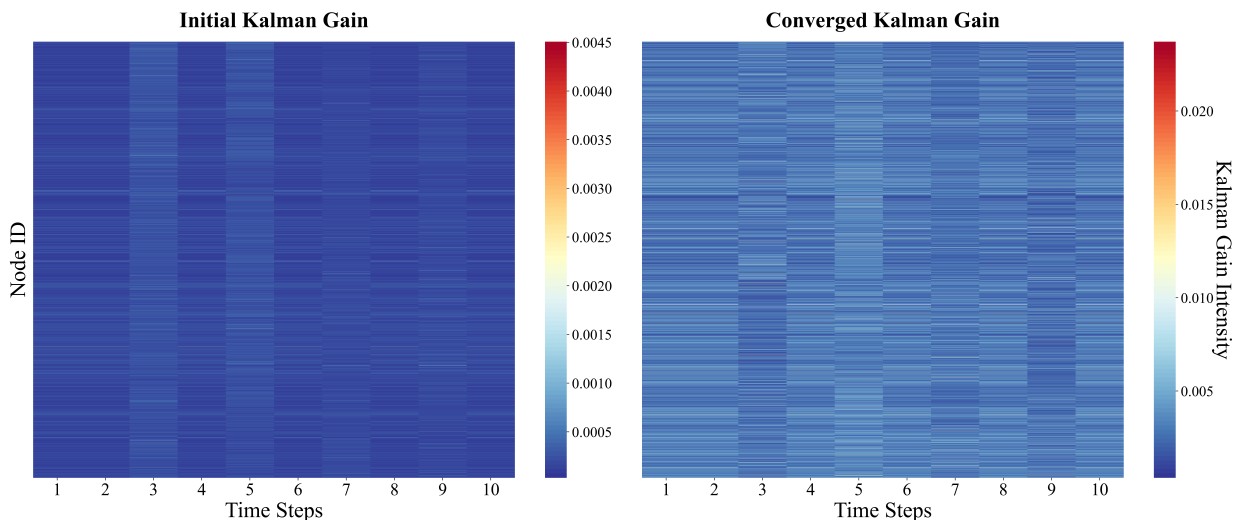

*Figure 10.* Spatiotemporal evolution of the Kalman gain on COLLAB (Initial vs. Converged).

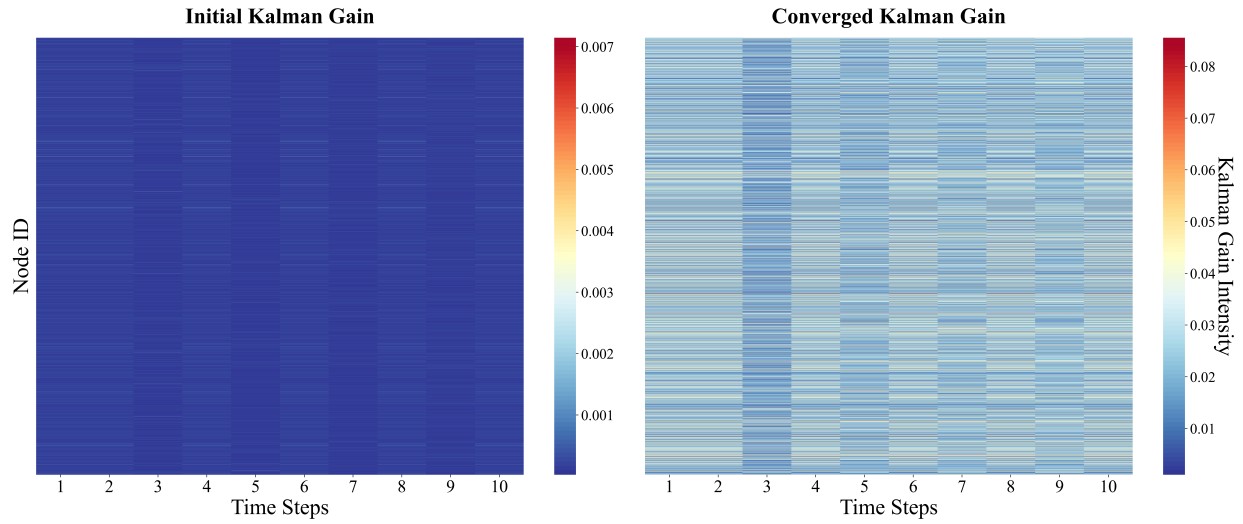

*Figure 11.* Spatiotemporal evolution of the Kalman gain on Yelp (Initial vs. Converged).

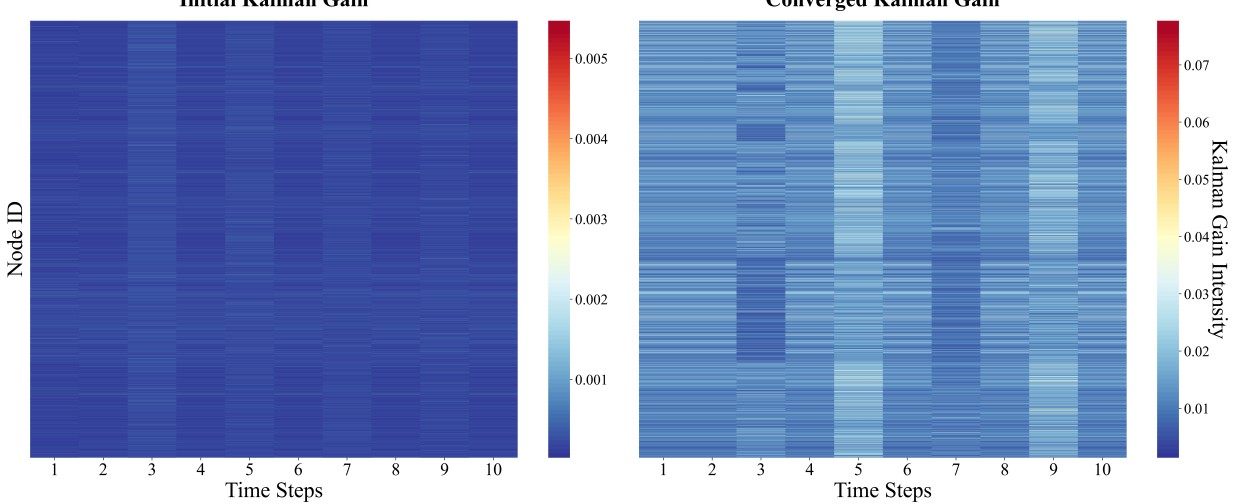

*Figure 12.* Spatiotemporal evolution of the Kalman gain on ACT (Initial vs. Converged).

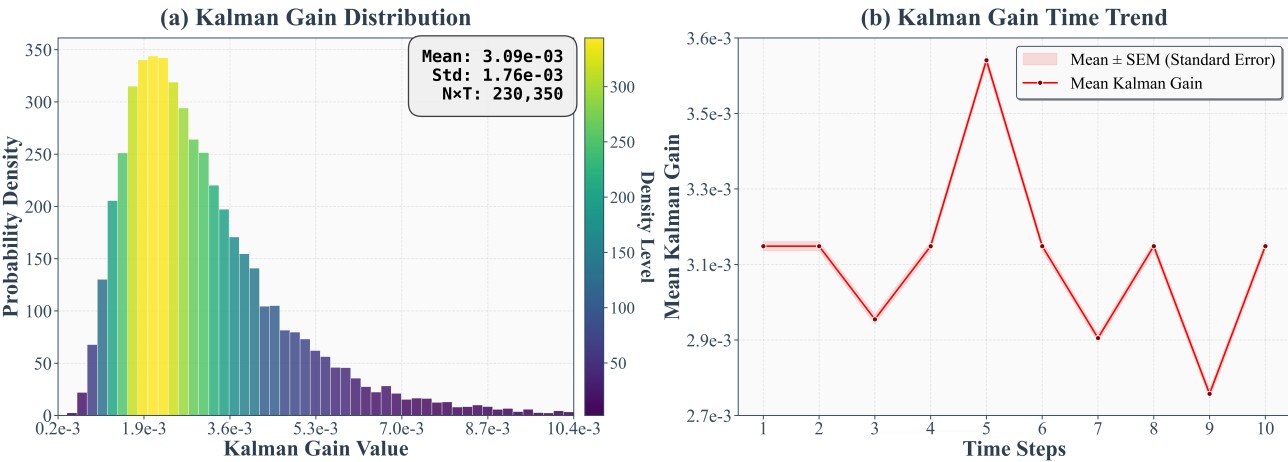

*Figure 13.* Empirical distribution and temporal dynamics of Kalman gain on COLLAB at convergence.

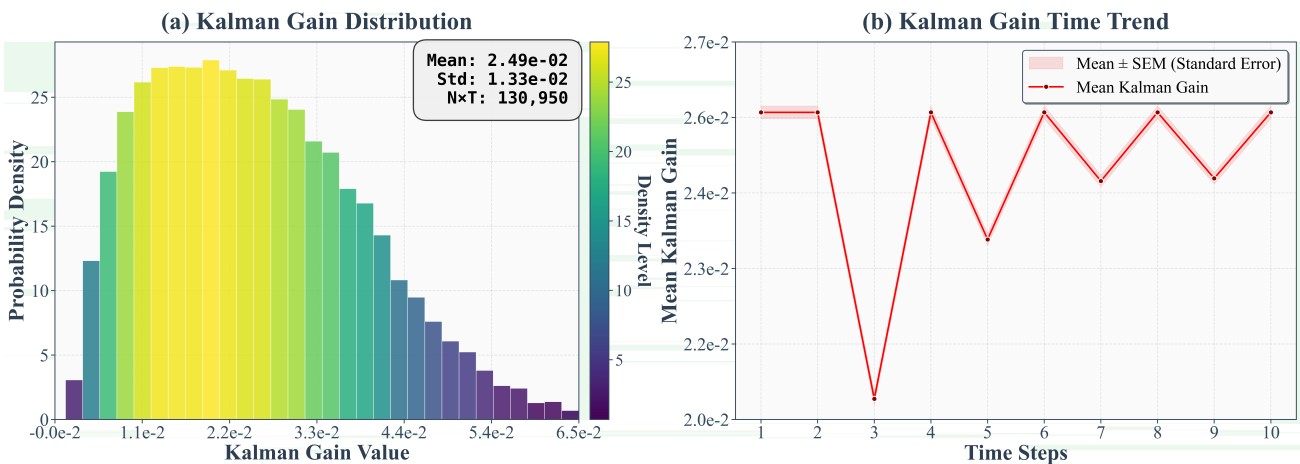

*Figure 14.* Empirical distribution and temporal dynamics of Kalman gain on Yelp at convergence.

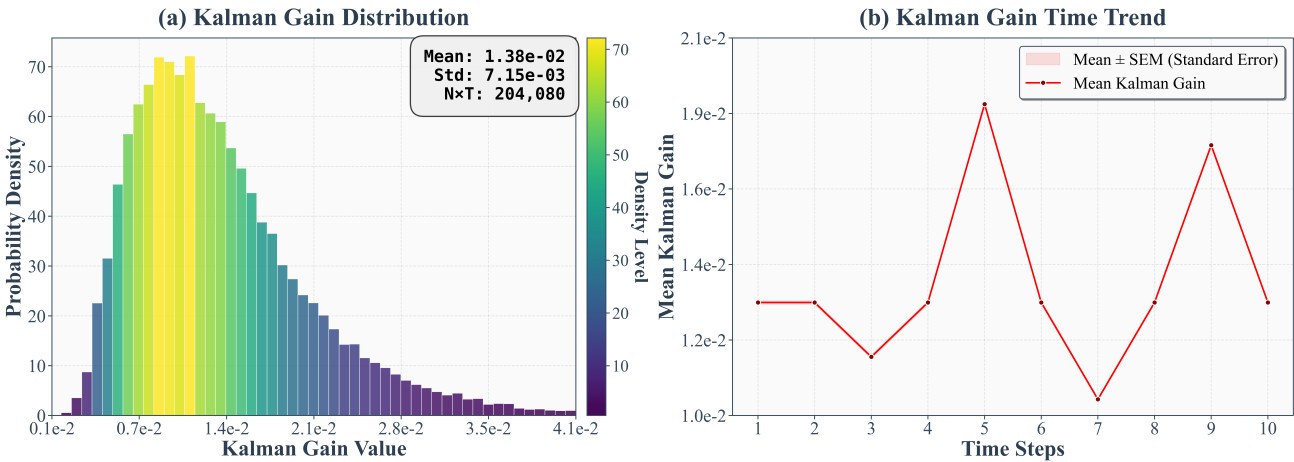

*Figure 15.* Empirical distribution and temporal dynamics of Kalman gain on ACT at convergence.

### F.2.2. SPECTRAL SIGNAL PURIFICATION AND FEATURE MAP EVOLUTION IN AFDM

In this section, we examine the progressive decoupling capability of the Adversarial-aware Frequency Decoupling Module (AFDM) throughout the training process. Figures 16–18 visualize the evolution of node feature maps. Structurally, the **rows** correspond to three distinct training stages (Initial, Middle, and Converged), while the columns display the feature states: the spatially-decoupled input (**Before**), the AFDM-modulated output (**After**), and the residual difference (**Difference**, calculated as $|\mathbf{X}_{\text{after}} - \mathbf{X}_{\text{before}}|$).

To quantify the intensity of the suppressed signals, we define the Perturbation Energy. Given that AFDM acts as a spectral filter designed to attenuate non-stationary noise, the residual difference between the input and output effectively proxies the magnitude of the removed perturbations. Thus, we formulate the energy as the mean absolute difference:

$$\text{Energy} = \frac{1}{N \times T} \sum_{i=1}^{N} \sum_{j=1}^{T} |x_{\text{after}}[i,j] - x_{\text{before}}[i,j]|. \qquad (47)$$

Since AFDM is designed to filter out non-stationary noise while preserving intrinsic dynamics, this energy metric effectively proxies the magnitude of the removed spectral perturbations.

**Visual analysis. (1) Horizontally**, as observed in the "Before" columns (left), the raw feature maps often exhibit chaotic, high-frequency fluctuations (e.g., sharp vertical stripes representing instantaneous disturbances). In contrast, the "After" columns (center) reveal significantly smoother temporal patterns, indicating that the intrinsic evolutionary trends are

preserved. Crucially, the "Difference" maps (right) capture the signal components removed by the module. Visually, the high-magnitude regions in these residuals exhibit a clear spatiotemporal correspondence with the abrupt vertical stripes in the input. This alignment demonstrates that the AFDM module effectively isolates and suppresses patterns characteristic of non-stationary noise, without discarding the intrinsic evolutionary trends. **(2) Vertically**, examining the evolution across rows (from initialization to convergence) reveals the progressive nature of the spectral purification process. In the early training stages, the "Difference" maps exhibit high perturbation intensity (quantified by elevated Energy values, e.g., 0.4595 on COLLAB), reflecting aggressive filtering to correct gross feature instabilities. As the model approaches convergence, the magnitude of these residuals significantly decreases (dropping to 0.0795), indicating that the AFDM transitions from aggressive correction to the fine-grained preservation of intrinsic dynamics. This quantitative decay confirms that the module adaptively minimizes its intervention to protect the robust representations learned by the model.

**Quantitative verification.** To quantitatively verify the convergence of the spectral purification process, we analyze the dynamics of the Perturbation Energy over training epochs, as presented in Figures 19–21. As visualized by the dashed trend lines, the energy exhibits a consistent diminishing trajectory across all datasets. Specifically, the negative slope values reported in the legends quantify the average rate of perturbation decay, providing a direct metric for the efficiency of the spectral purification. Crucially, the actual energy curves flatten in the later stages, indicating that the system stabilizes over time. Initially, AFDM performs aggressive filtering (indicated by high energy values) to suppress severe high-frequency spectral noise. As training converges, the module transitions to a fine-grained modulation regime. This quantitative decay confirms the effectiveness of our entropy-regularized frequency modulation in systematically purifying harmful signals, thereby ensuring robust feature inputs for downstream modeling.

## F.3. Additional Analysis: Robustness to Non-Targeted Adversarial Attacks

This section provides supplementary analysis to further explain the results presented in Section 5.2.

- **Consistent performance superiority.** Across the vast majority of scenarios, including both the original clean datasets and those subjected to structure or feature attacks, DeR-Mamba consistently outperforms all competing baselines. Specifically, on the COLLAB dataset, DeR-Mamba achieves an AUC of 93.74% under structure attacks. It also demonstrates substantial improvements over the runner-up (DG-Mamba), achieving performance gains of 1.14% under structure attacks, and 1.63%, 1.11%, and 0.55% under feature attacks with noise intensities of $\gamma \in \{0.5, 1.0, 1.5\}$, respectively. A similar trend is observed on the Yelp dataset, where our method outperforms the second-best baseline with margins of 0.46%, 1.42%, 2.07%, and 3.27% across the corresponding structure and feature attacks. Notably, these performance gaps are even more pronounced when compared to other robust baselines such as RGCN, WinGNN, and DGIB-Bern. This consistent superiority validates the effectiveness of our multi-domain decoupling framework in maintaining high predictive accuracy even when the underlying graph data is compromised by compound perturbations.

- **Compared with static/dynamic GNNs and DGSL baselines.** Standard baselines struggle to adapt to these adversarial conditions due to their inherent architectural limitations. Static GNNs fail to capture spatiotemporal evolution, making them unsuitable for dynamic scenarios where structure and features shift over time. Dynamic GNNs (e.g., DySAT, SpoT-Mamba) perform sub-optimally because, while they model temporal dependencies, they typically neglect the accumulation of residual perturbations during the evolution process. This oversight leads to the reinforcement of invalid or misleading structural information as the graph evolves. Similarly, Dynamic Graph Structure Learning (DGSL) methods (e.g., TGSL) lack explicit mechanisms to handle uncertainty induced by structural incompleteness, making them highly sensitive to structural perturbations. Under structure attacks, RDGSL and TGSL both suffer performance drops exceeding 10%, decreasing from 82.29% to 71.36% and from 84.09% to 73.66%, respectively.

- **Compared with Robustness-based baselines.** While Robust DGNNs are explicitly designed to enhance stability in noisy environments, they are generally limited by a single-view perspective and lack a systematic approach to decoupling multi-dimensional perturbations. For instance, DGIB utilizes the Information Bottleneck principle to suppress noise via information compression. However, due to its heavy reliance on the Hidden Markov assumption, it struggles to capture long-range dependencies and cannot simultaneously address complex, compound disturbances, resulting in unremarkable performance in most scenarios. Similarly, DG-Mamba mitigates structural noise to some extent through state-space modeling but remains confined to the structural domain, failing to address perturbations in the spectral domain. This limitation restricts their robustness when facing multi-source compound interference.

In summary, DeR-Mamba achieves competitive robustness by decoupling spatial, spectral, and temporal dynamics. Unlike baselines relying on single-source denoising, it decouples multi-domain entanglement, ensuring superior performance.

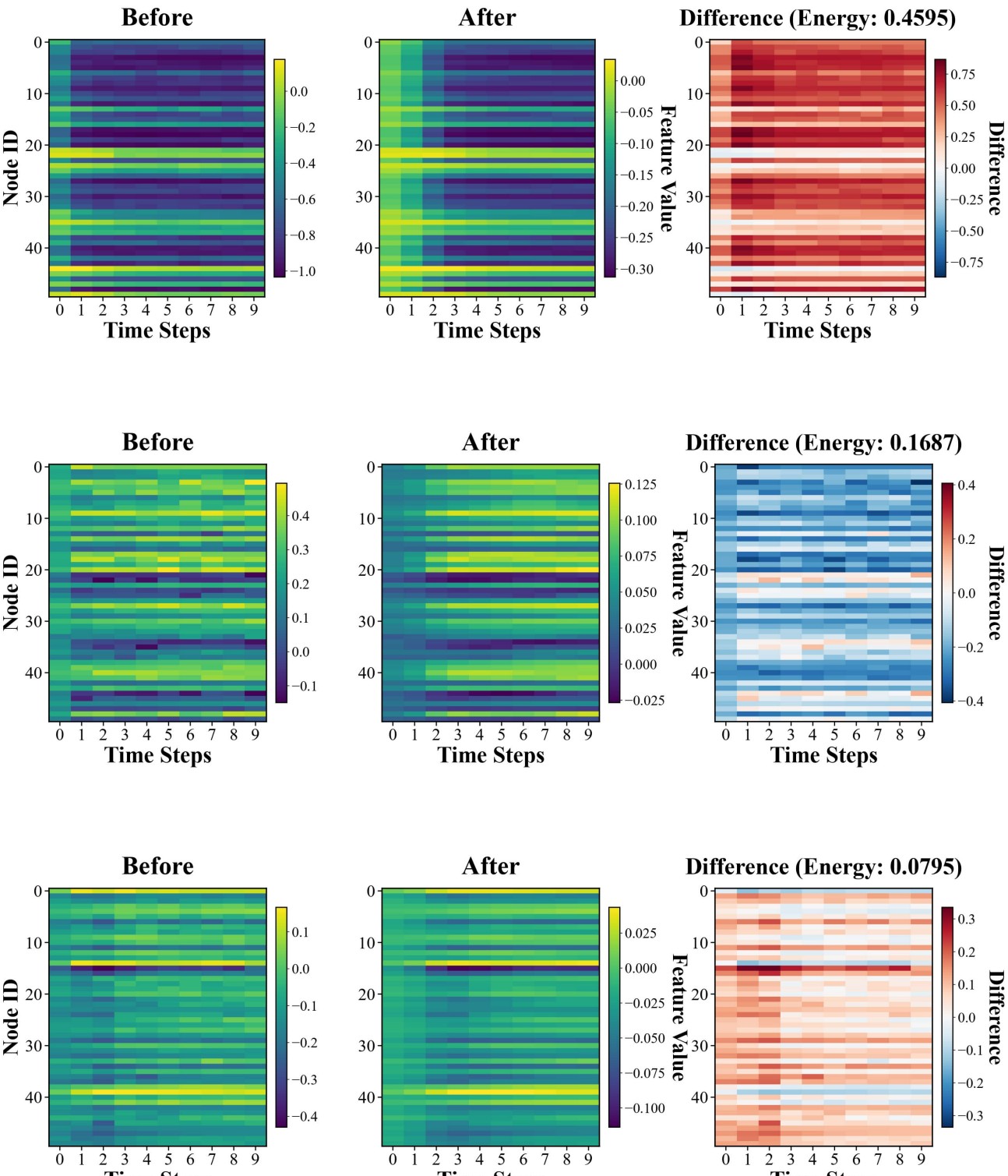

*Figure 16.* Temporal evolution of node feature maps during training on COLLAB, illustrating the progressive decoupling of non-stationary frequency components by AFDM from initialization to convergence (top to bottom).

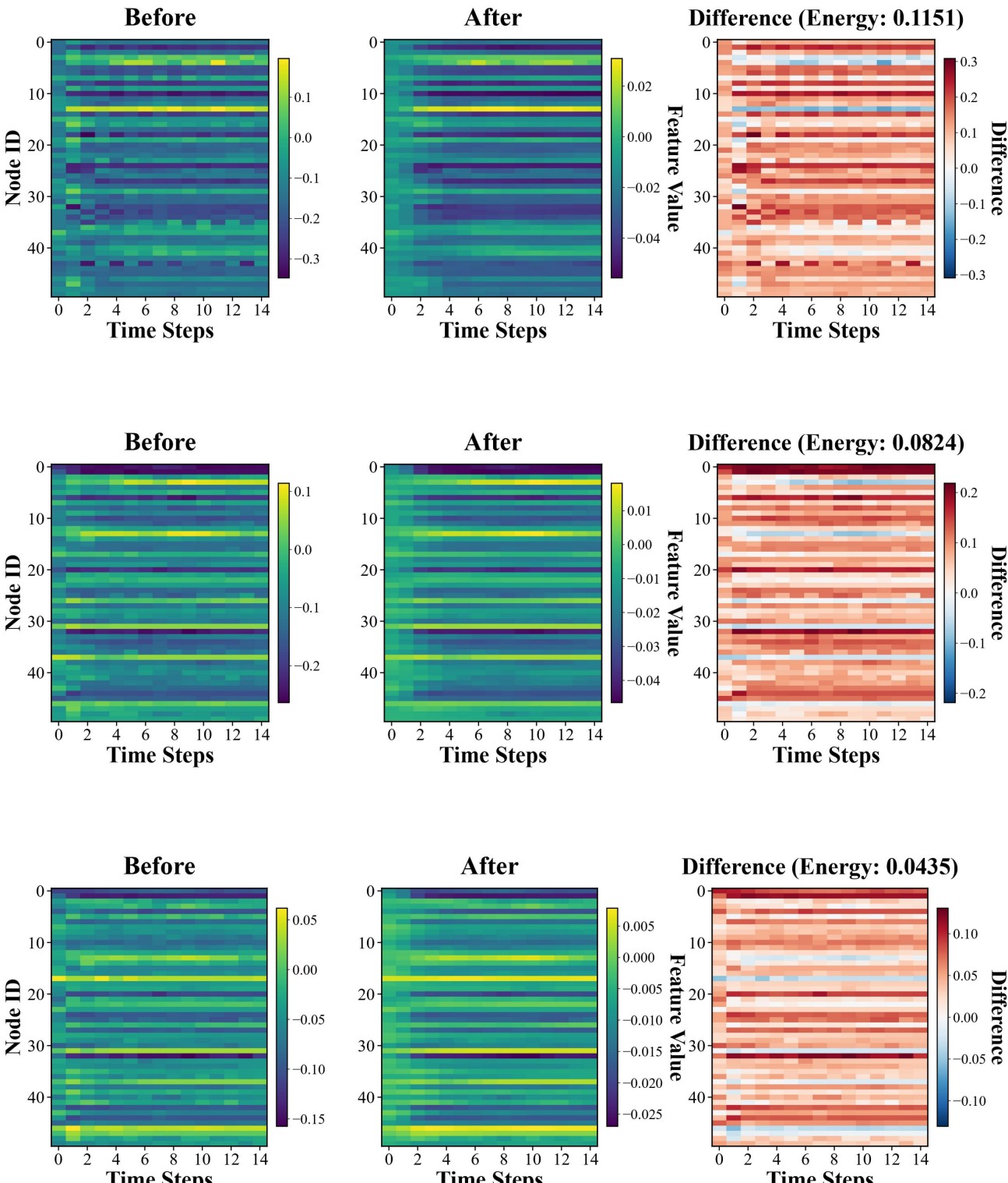

*Figure 17.* Temporal evolution of node feature maps during training on Yelp, illustrating the progressive decoupling of non-stationary frequency components by AFDM from initialization to convergence (top to bottom).

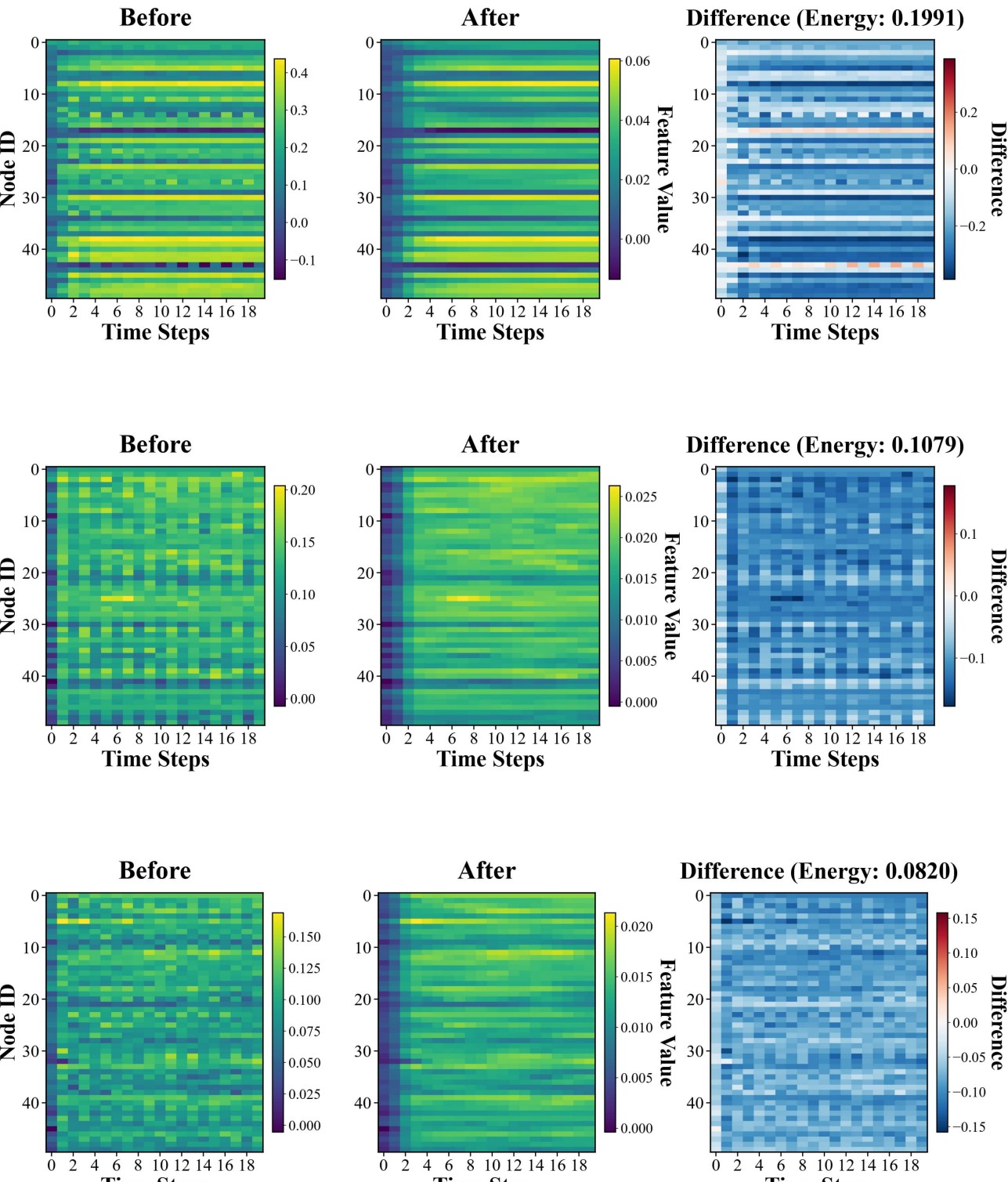

*Figure 18.* Temporal evolution of node feature maps during training on ACT, illustrating the progressive decoupling of non-stationary frequency components by AFDM from initialization to convergence (top to bottom).

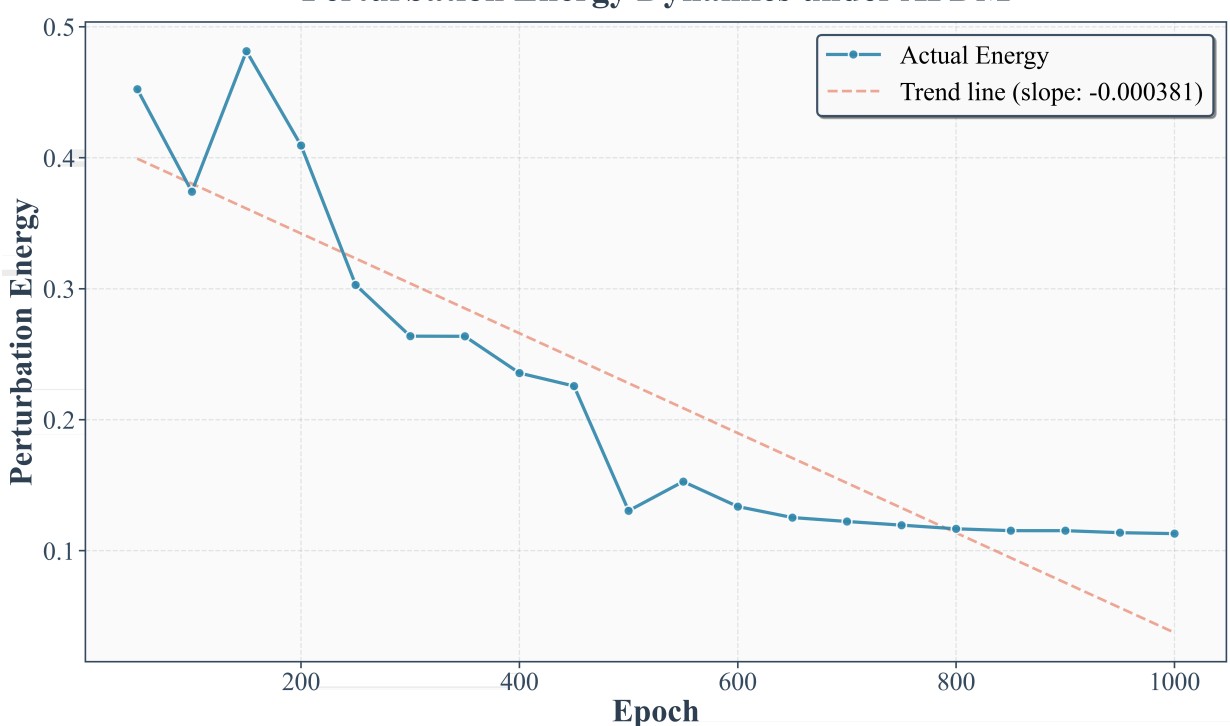

*Figure 19.* Perturbation energy dynamics under AFDM on COLLAB.

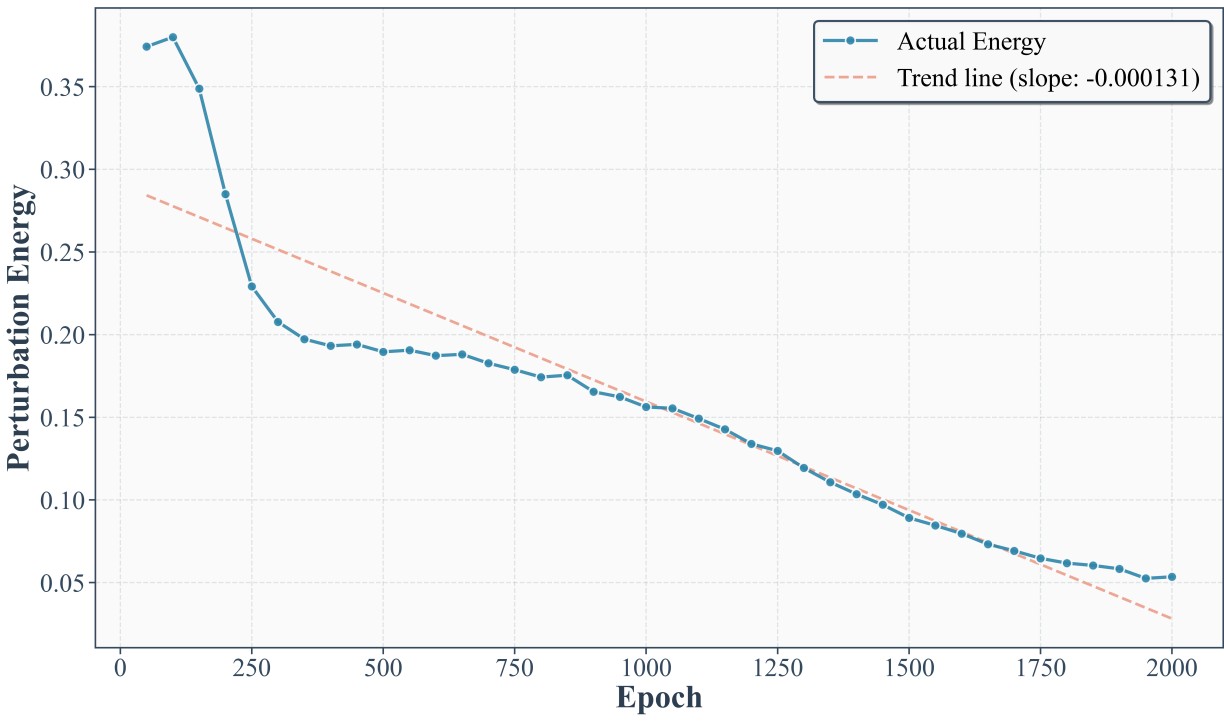

*Figure 20.* Perturbation energy dynamics under AFDM on Yelp.

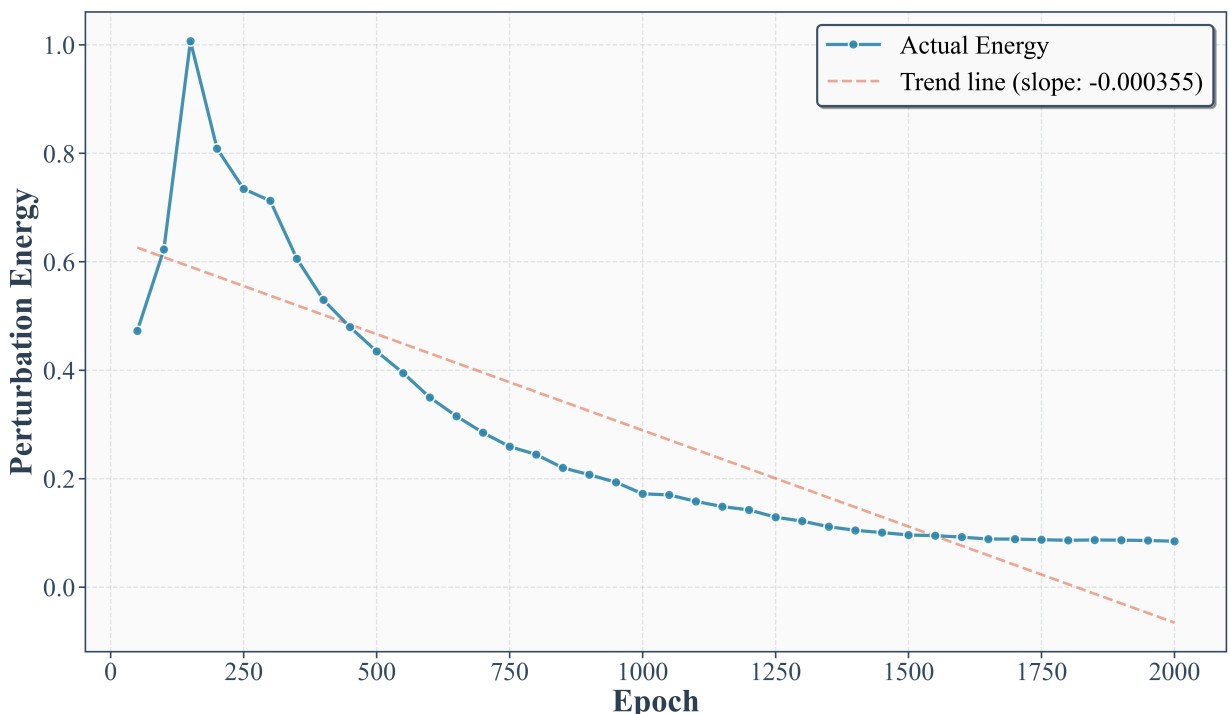

*Figure 21.* Perturbation energy dynamics under AFDM on ACT.

## F.4. Additional Analysis: Robustness to Targeted Adversarial Attacks

This section provides supplementary analysis to further elucidate the results presented in Section 5.3. Compared to non-targeted scenarios, targeted attacks present a more rigorous challenge, as adversaries can exploit specific fragility patterns within the dynamic graph to mislead model predictions. In this section, we compare the performance of DeR-Mamba against the top-performing baselines from Table 1 under two targeted adversarial settings: **evasion attacks** and **poisoning attacks**. The results are detailed in Table 2. We report the relative performance degradation ($\Delta\%_\downarrow$) induced by these attacks, calculated as:

$$\Delta\%_\downarrow = \frac{\text{AUC}_{\text{Original}} - \text{AUC}_{\text{Attack}}}{\text{AUC}_{\text{Original}}} \times 100\%. \tag{48}$$

- **Superior stability against targeted perturbations.** Compared to other baselines, DeR-Mamba exhibits the minimal average performance degradation (Avg. $\Delta\%_\downarrow$) across almost all evasion and poisoning attack scenarios. For instance, on the Yelp dataset, DeR-Mamba limits the average performance drop to just 13.9% for evasion attacks and 13.4% for poisoning attacks. In contrast, the second-best method, DG-Mamba, suffers significantly higher degradation of 22.1% and 21.3%, respectively. This empirical evidence highlights the significant robustness advantage of our framework in preserving predictive integrity under targeted malice.

- **Compared with static/dynamic GNNs and DGSL baselines.** Standard baselines generally display marked vulnerability under targeted attacks. For example, Static and Dynamic GNNs struggle to withstand structural manipulation. Specifically, SpoT-Mamba experiences a severe AUC drop of 15.3% in the evasion attack setting ($n_p = 1$) on Yelp, representing the largest decline among all baselines. Similarly, DGSL methods like TGSL improve resilience by learning substructure variations, they still encounter severe performance deterioration. A clear instance is observed in the Yelp dataset under poisoning attacks ($n_p = 2$): TGSL's AUC plummets from 76.55% to 56.27%, whereas DeR-Mamba demonstrates far greater stability, declining only from 84.46% to 76.67%. This underscores that relying solely on temporal evolution or sub-structure modeling is inadequate for coping with complex adversarial perturbations.

- **Comparison with robustness-focused baselines.** Robust DGNNs such as WinGNN, DGIB, and DG-Mamba exhibit relative stability. However, these methods lack a systematic decoupling of multi-dimensional noise. Consequently,

as the attack intensity ($n_p$) increases, their AUC scores still suffer precipitous declines, failing to maintain consistent defense levels. For instance, on the ACT dataset under poisoning attacks with $n_p = 3$, DGIB-Cat experiences a drastic performance drop from 94.89% to 64.38%, representing a 32.2% degradation that is among the most severe across all baselines. In contrast, DeR-Mamba demonstrates superior resilience by declining only 18.7% from 97.17% to 79.02%. This significant disparity underscores the critical necessity of multi-domain decoupling in preserving predictive integrity against intense adversarial perturbations.

In summary, DeR-Mamba achieves superior robustness through a multi-domain decoupling strategy. Specifically, MP–K$^2$alman samples latent evolutionary pathways to construct a posterior approximation of the perturbed structure. AFDM suppresses the spatiotemporal propagation of non-stationary high-frequency perturbations while reinforcing stationary low-frequency signals. In addition, the selective state-space mechanism captures global intrinsic dynamics and constrains the accumulation of long-sequence residual noise. This systematic decoupling across spatial, spectral, and temporal domains enables DeR-Mamba to significantly outperform all baselines under both non-targeted and targeted adversarial attacks.

### F.5. Additional Results: Hyperparameter Sensitivity Analysis

We provide additional hyperparameter sensitivity analyses in Figures. 22–25 under the Original, Feature Attack, Evasion Attack, and Poisoning Attack settings, based on which a more comprehensive investigation is conducted. From the resulting three-dimensional response surfaces, several conclusions can be drawn as follows.

● **Impact of $\beta_1$ on temporal redundancy suppression.** The hyperparameter $\beta_1$ controls the strength of cross-temporal redundancy suppression. As observed on the COLLAB, Yelp, and ACT datasets, moderately increasing $\beta_1$ effectively alleviates the impact of accumulated redundancy and leads to consistent performance improvements, where the AUC often reaches its peak. This indicates that a proper redundancy regularization term enables the model to capture more stable temporal structures. In contrast, excessively large values of $\beta_1$ over-constrain the information flow, resulting in degraded performance.

● **Impact of $\beta_2$ on the Shrinkage–KL trade-off.** The hyperparameter $\beta_2$ balances the reconstruction-related term and the KL divergence regularization. Experimental results demonstrate that an appropriate $\beta_2$ preserves distributional diversity while enforcing consistency between $\mathbf{Z}_{\text{Seq}}$ and $\bar{\mathbf{Z}}_{\text{Dec}}$, thereby mitigating the progressive accumulation of redundancy during temporal propagation. When $\beta_2$ lies within a reasonable range, the model learns more discriminative dynamic representations for temporal sequence modeling. However, overly large $\beta_2$ excessively amplifies the consistency constraint, limiting representational flexibility and ultimately leading to a decline in AUC performance.

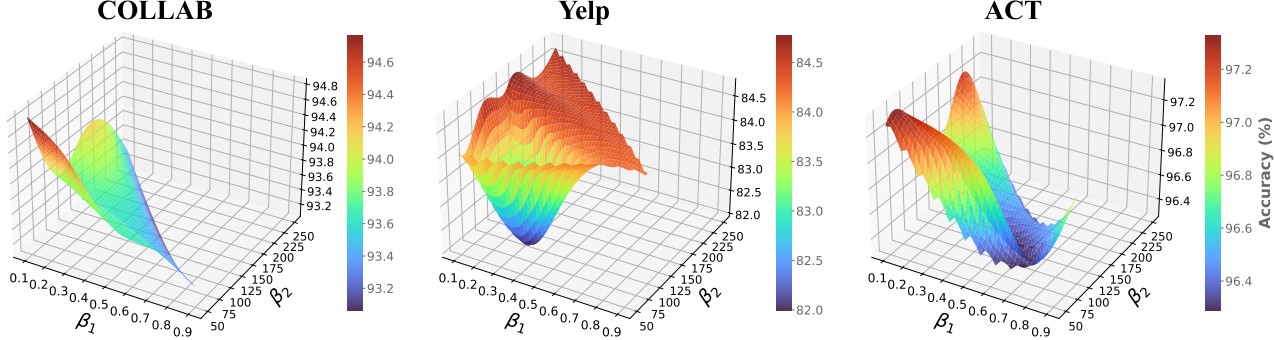

*Figure 22.* Hyperparameter sensitivity analysis under Original settings.

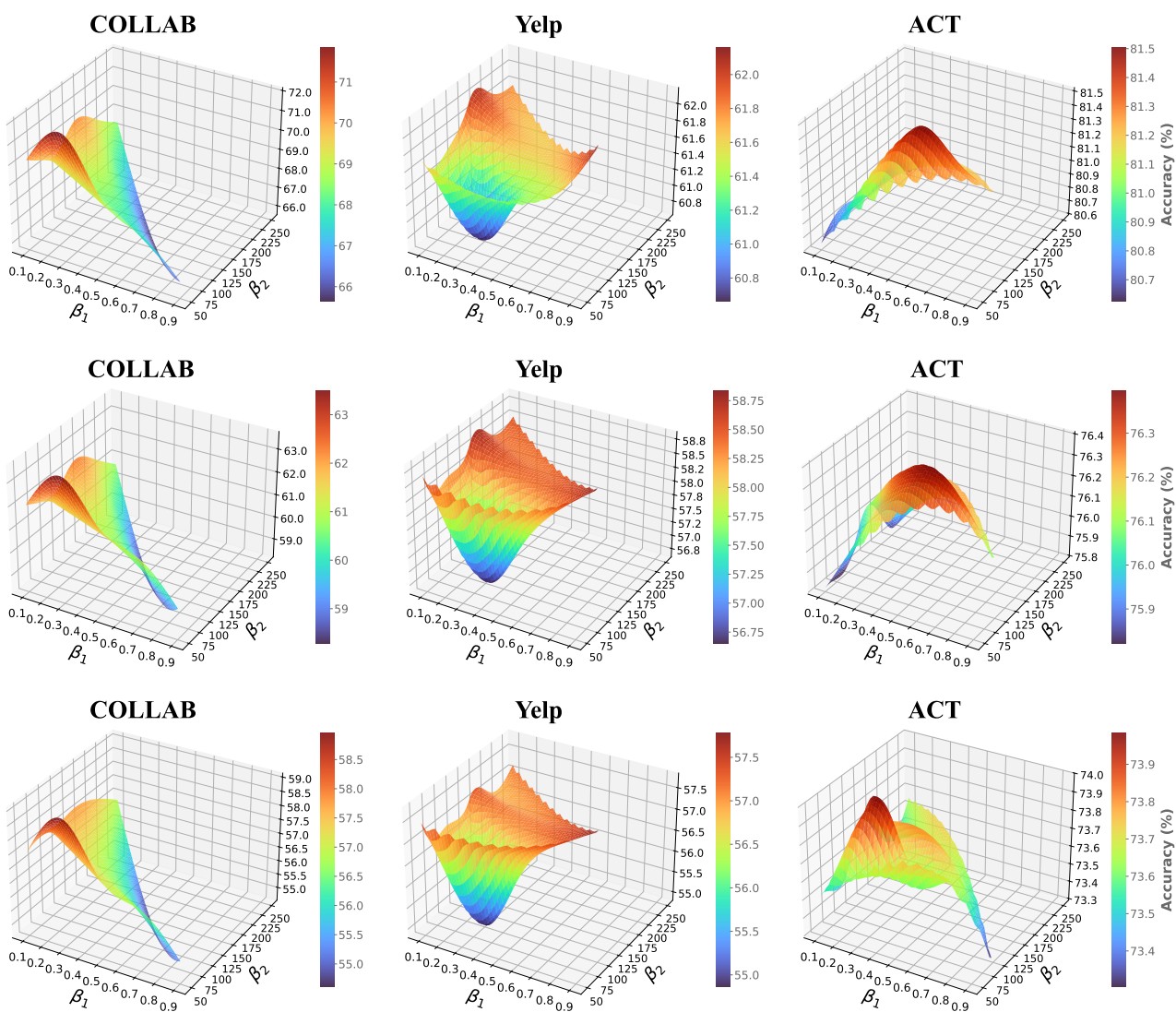

*Figure 23.* Hyperparameter sensitivity analysis under feature attack settings with $\gamma \in \{0.5, 1.0, 1.5\}$ (from top to bottom).

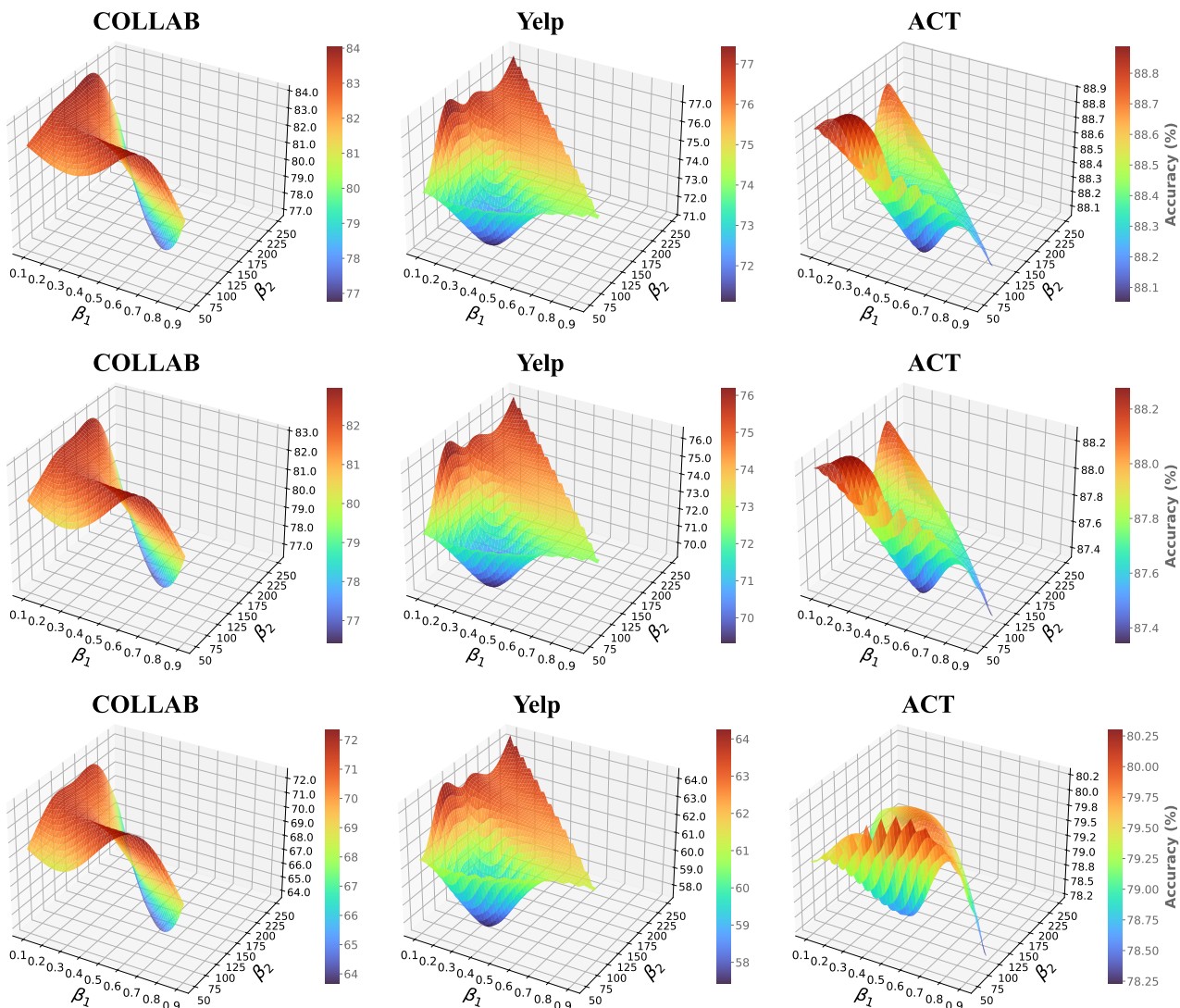

*Figure 24.* Hyperparameter sensitivity analysis under evasion attack settings with $n_p \in \{1, 2, 3\}$ (from top to bottom).

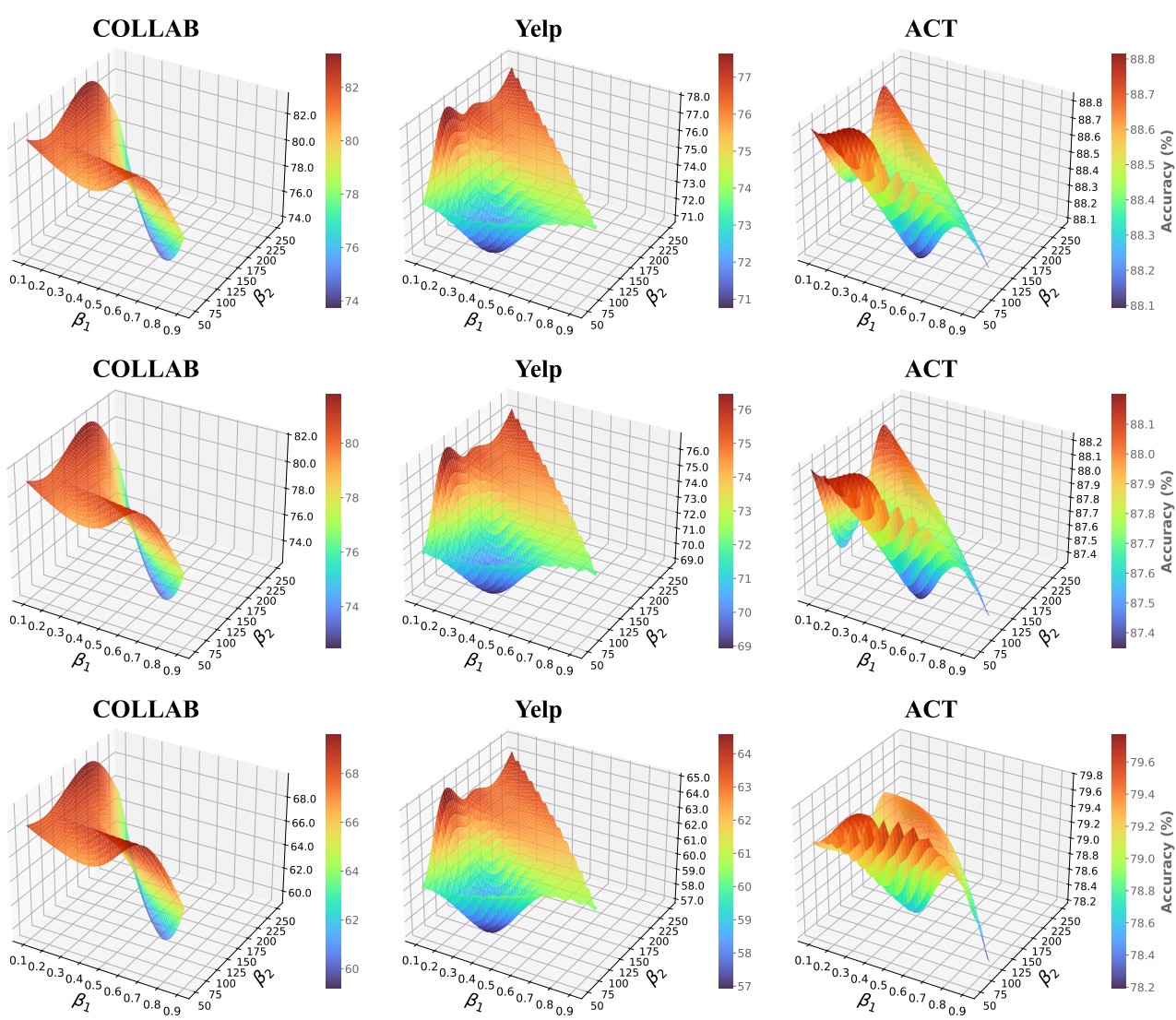

*Figure 25.* Hyperparameter sensitivity analysis under poisoning attack settings with $n_p \in \{1, 2, 3\}$ (from top to bottom).

# G. Parameter Settings

For fair comparison, we follow the standard configurations commonly used in the original baseline papers, setting the number of GNN layers to 2 for all baselines and to 1 for DeR-Mamba, as a single layer is sufficient to perform multi-domain decoupling while maintaining strong robustness. The latent dimension is fixed to 128. For each baseline method, we adhere to the hyperparameter recommendations in their original papers and further tune them for fairness. For DeR-Mamba, we provide candidate hyperparameter ranges in the configuration file and perform grid search on the validation set to identify the optimal combination. Specifically, we use the Adam optimizer (Kingma & Ba, 2015), with the learning rate chosen from $\{1e-1, 1e-2, 1e-3, 2.5e-3, 5e-3, 1e-4, 1e-5\}$, dropout selected from $[0.0, 0.3]$, and weight decay from $\{1e-5, 5e-5, 1e-4, 5e-4, 1e-3\}$. The maximum number of training epochs is 2000. The search ranges for $\beta_1$ and $\beta_2$ are $[0.1, 0.9]$ and $\{50, 100, 150, 200, 250\}$, respectively. The number of particles $M$ is selected from $\{1, 3, 5, 7, 9\}$. Early stopping is adopted to prevent overfitting based on the validation performance.

# H. Ablation Studies

We further provide a more detailed ablation analysis. We perform ablation studies to assess the contribution of each component in DeR-Mamba. MP–K$^2$alman and AFDM are progressively removed, after which the model is evaluated under both non-targeted and targeted adversarial attacks. For feature, evasion, and poisoning attacks, we use the most challenging COLLAB dataset to highlight the contribution of each module. Results are reported in Table 3 and Figure 3.

**Impact analysis of MP–K$^2$alman.** When MP–K$^2$alman is removed, model performance on the original setting drops by 3.04%, 2.73%, and 1.82% across the three metrics. Under structure attacks, performance decreases by 3.14%, 3.87%, and 1.90%. These results reveal severe degradation when the model faces structural perturbations. This demonstrates that multi-particle sampling and Kalman-style updates play a key role in modeling structural uncertainty, and are crucial for robustness against incomplete or corrupted graph structures. Under feature, evasion, and poisoning attacks, the degradation further intensifies as attack strength increases, confirming the necessity of MP–K$^2$alman under multi-domain entanglement.

**Impact analysis of AFDM.** Removing AFDM also reduces performance on the original data and under structure and feature attacks. Under targeted attacks, the degradation becomes significantly larger. As shown in Table 3, when attack strength increases ($n_p = 1, 2, 3$), the performance drops by 4.36%, 4.19%, 4.40%, and 4.60%, 4.57%, 3.95%, respectively. Moreover, the relative $\Delta\%$ under targeted attacks is clearly larger than under non-targeted settings. These results show that AFDM is essential for defense against complex perturbations. AFDM separates unstable high-frequency components from stable low-frequency components, preventing harmful high-frequency signals from propagating through time and improving robustness by strengthening frequency-domain discrimination.

**Impact analysis of Joint Effects.** The complete DeR-Mamba model exhibits strong robustness across all attack settings, including both non-targeted and targeted adversarial attacks. Notably, under evasion and poisoning attacks in Table 2, the average performance degradation (Avg. $\Delta\%_{\downarrow}$) is only 18.0% and 18.5%, much lower than most baselines. This advantage arises because DeR-Mamba simultaneously addresses structural incompleteness and frequency perturbations while incorporating selective state-space modeling to suppress temporal error accumulation.

# I. Related Work

## I.1. Robust Dynamic Graph Learning

Dynamic Graph Neural Networks (DGNNs) aim to establish or update temporal dependencies between nodes by capturing the time-varying information required to refresh node embeddings, thereby enabling models to track the evolution of graph structures and node states. Existing DGNNs typically model temporal dynamics by extracting spatial or temporal features locally. For example, SimpleDyG (Wu et al., 2024) transforms spatial graph convolution into a node-wise sequential modeling paradigm and employs Transformer architectures to capture spatiotemporal interactions within dynamic graphs. However, DGNNs are prone to over-smoothing, which can lead to unstable representations, reduced robustness, and increased susceptibility to adversarial structural perturbations. To mitigate this issue, adversarial dynamic graph defense models have been explored, among which ADGNN (Li et al., 2024a) injects perturbations during training to enhance node-level adversarial robustness, while T-SHIELD (Lee et al., 2024) adopts an adversarial training framework to alleviate the impact of structural attacks on dynamic graphs. Nevertheless, these DGNNs generally assume that the observed graph at each time step is relatively "clean," even though their outputs exhibit a certain degree of robustness. In real-world scenarios,

dynamic graphs often involve various types of noise and dynamic disturbances unrelated to prediction objectives, which can gradually degrade edge reliability and weaken the integrity of node representations. To address the degradation of graph edge quality under dynamic and noisy conditions, BandRank (Li et al., 2025b) introduces a frequency-based dynamic graph learning strategy that ranks candidate edges according to their spectral characteristics and prioritizes meaningful temporal interactions. Similarly, Dynamic Graph Information Bottleneck (DGIB) (Yuan et al., 2024) seeks robust representations that preserve task-relevant information while suppressing perturbation-sensitive components. In addition, wavelet-based GNNs (Hammond et al., 2011) have attracted significant attention for enhancing graph stability, among which Graph Wavelet Neural Networks (GWNN) (Xu et al., 2019) integrate spectral graph wavelets into spatial GNNs to capture both local and global structural properties. Building on this line of work, SEA-GWNN (Deb et al., 2024) extends wavelet-based modeling to higher-frequency domains while preserving local spectral information, although it remains limited to static graph settings. More recently, learnable dynamic graph wavelet mechanisms (Bastos et al., 2023) have been proposed to adaptively adjust wavelet scales over time. Although these wavelet-based approaches help alleviate instability caused by dynamic noise or structural changes, they still rely primarily on single-scale or global spectral features, making it challenging to capture the interplay between stable low-frequency signals and fluctuating high-frequency components in real-world dynamic graphs with multi-scale interactions and cross-temporal dependencies.

Furthermore, dynamic graph perturbations in practical environments arise from multiple sources. For example, structural incompleteness, spectral distortion, and historical noise accumulation. The aforementioned methods focus on modeling perturbations from a single perspective and therefore cannot adequately address the multi-source, multi-domain disturbances encountered in real-world dynamic graphs. To the best of our knowledge, no existing work provides a unified robust dynamic graph learning framework that systematically handles multi-domain perturbations from spatial, spectral, and temporal perspectives. In this paper, we propose DeR-Mamba, which decomposes and suppresses perturbations across these three domains and achieves stable representation learning under complex multi-source disturbances.

### I.2. Kalman Filtering in Graph-Structured Systems

The Kalman filter is a linear, unbiased, minimum-variance recursive estimator capable of performing optimal state estimation for dynamical systems under incomplete and noisy observations. Its prediction–correction mechanism and uncertainty-aware fusion make it highly effective in applications, such as attitude control (Cha et al., 2019) and multi-target tracking under sensor noise (Gustafsson et al., 2002). These properties naturally align with the challenges in dynamic graphs, whose temporal evolution is often affected by structural incompleteness and noise contamination. During the prediction stage, the filter propagates system states using the transition model to obtain a prior estimate. In the correction stage, it fuses the prior with current observations to refine the estimate and reduce uncertainty. This fusion mechanism provides the minimum-variance estimate by optimally weighting predictive and observational uncertainties. Such capabilities make the Kalman filter particularly suitable for robustly estimating latent relationships among nodes under noisy, uncertain, and dynamically evolving graph structures. Several recent studies have begun exploring Kalman-inspired methods for graph-based state estimation. A hybrid optimization framework (Buchnik et al., 2024) that integrates gradient-driven and data-driven filtering has been proposed. GSP-KalmanNet (Buchnik et al., 2024) was developed to address state estimation on graphs with complex signal dynamics, while a parallel Kalman filtering formulation (PKF) (Liu et al., 2024) was further introduced to reduce the computational burden of classical Kalman updates in multi-target tracking.

To date, however, the application of Kalman filtering to dynamic graphs remains largely unexplored, let alone its use as a means to improve robustness. The key challenges arise from the high-dimensional and irregular graph topology, the spatial dependencies embedded in node representations, and the difficulty of estimating low-dimensional latent states under diverse and interacting perturbations. Moreover, dynamic graph perturbations often stem from multiple heterogeneous sources, making classical Kalman filtering architectures difficult to directly apply or adapt. Our proposed MP–K$^2$alman addresses these issues by employing a multi-particle representation to capture structural uncertainty through kernelized embeddings, together with a Kalman-style state update that supports confidence-weighted correction under multi-domain entanglement. This design enables explicit modeling of structural uncertainty throughout spatiotemporal evolution, mitigates estimation drift, and fills an important methodological gap in robust dynamic graph learning.

### I.3. Graph Modeling with State Space Models

State Space Models (SSMs) have played a central role in dynamic system modeling since early control theory (Aoki, 2013), with broad applications across scientific and engineering domains. In modern deep learning, SSMs are recognized for their efficiency in capturing long-range temporal dependencies. The core idea of SSMs is to leverage linear time-invariant system

dynamics to map input sequences to output sequences, thereby modeling structural patterns that unfold over time. To enable differentiable learning, discrete-time SSMs introduce a learnable step-size parameter (Gu et al., 2020), allowing end-to-end training via gradient-based optimization. Building on this formulation, Structured State Space Models (S4) further improve computational efficiency and scalability through reparameterization techniques.

Recently, the Mamba architecture (Gu et al., 2020) has demonstrated strong performance in long-sequence modeling, motivating efforts to extend selective state-space mechanisms to graph-structured data. Several studies attempt to reformat non-Euclidean structures into sequence-like forms to make SSMs applicable to graph learning tasks (Behrouz & Hashemi, 2024; Wang et al., 2025b; Hu et al., 2025). However, graph-specific challenges, such as node-ordering ambiguity and the inherent non-stationarity of evolving graph states, complicate this adaptation. Directly applying sequence models to graph signals overlooks critical spatial relationships, limiting their ability to capture graph-dependent long-range interactions and global structural context. To mitigate these issues, SpoT-Mamba (Choi et al., 2024) is proposed as a spatiotemporal forecasting framework that generates embeddings by scanning multiple random-walk sequences over the graph. DG-Mamba (Yuan et al., 2025) was further developed as a dynamic graph learning approach that integrates selective SSMs with adaptive structural updates, enhancing the model's ability to capture global interaction patterns. Despite these advances, sequence-structured SSMs and their variants remain constrained when applied to dynamically evolving non-Euclidean systems. Dynamic graphs exhibit node-order uncertainty, non-stationary state transitions, and compounding perturbations across structural, temporal, and feature dimensions. These factors accumulate and propagate through time, exacerbating the modeling difficulty of selective state-space architectures such as Mamba, and introducing unresolved challenges in state selection, information fidelity, and robustness—areas that have yet to receive systematic investigation. Our proposed DeR-Mamba addresses these limitations through a principled perturbation-decoupling mechanism and a robust dynamic state modeling strategy. By jointly mitigating structural, frequency-domain, and state-evolution perturbations, DeR-Mamba substantially improves stability and generalization in complex dynamic graph environments.

## J. Implementation Details

### J.1. Hardware-Aware Dynamic Graph Redundancy-aware Scan Implementations

Modeling long-range dependencies in large-scale dynamic graphs typically incurs prohibitive computational and memory overhead. To mitigate these challenges, we implement DeR-Mamba by integrating a hardware-aware optimization strategy (Yuan et al., 2025) within the Dynamic-Graph Redundancy-aware Scan (DGRS), aimed at enhancing both systemic efficiency and scalability. We leverage hardware-specific architectural features to minimize memory footprints while maximizing computational throughput.

Specifically, DeR-Mamba fetches input data $\{\mathcal{A}, \mathcal{B}, \mathcal{C}, \Delta\}$ from High Bandwidth Memory (HBM) with a complexity of $\mathcal{O}(BTD + ND)$. The graph structure is subsequently discretized, and intermediate states—scaling at $\mathcal{O}(BTDN)$—are processed within the Static Random Access Memory (SRAM), with the final outputs re-coalesced into HBM at a complexity of $\mathcal{O}(BTD)$. This discretization step effectively isolates the primary evolution paths critical for long-range dependency modeling, thereby filtering out redundant noise and auxiliary temporal structures. Within the SRAM, we utilize parallel associative scans to compute intermediate states. By exploiting computational locality, the data can be processed at significantly higher speeds, reducing the frequency of large-scale memory access and substantially accelerating overall throughput. The proposed method reduces I/O overhead to $\mathcal{O}(N)$, effectively minimizing the number of memory read/write operations. By fusing multiple computational steps into a single optimized kernel, we not only decrease redundant memory invocations but also preserve the structural integrity of long-range dependencies across the dynamic graph. To accommodate ultra-long sequences that exceed SRAM capacity, we employ a chunk-based strategy, partitioning the sequences into sub-blocks where intermediate states are passed sequentially to maintain global dependency consistency. Furthermore, during the backpropagation phase, we utilize recomputation to avoid the $\mathcal{O}(BTDN)$ storage requirement for intermediate states. This strategy alleviates HBM memory pressure and mitigates the latency associated with frequent data transfers. Consequently, by integrating these hardware-aware optimizations, DeR-Mamba maintains robust long-range modeling capabilities while achieving superior efficiency and scalability for large-scale dynamic graph scenarios.

### J.2. Hardware and Software Configurations

The experimental environment is configured as follows:

- **Operating System:** Ubuntu 20.04 LTS

- **CPU:** 10 vCPUs, Intel Xeon Processor (Skylake, IBRS) with 72 GB memory

- **GPU:** NVIDIA A100 (PCIe) with 40 GB memory

- **Software:** CUDA 11.7, Python 3.9, PyTorch 1.12.1, PyTorch Geometric 2.0.1

## K. Limitations

The limitations of DeR-Mamba are primarily observed in the following aspects. First, while we have verified the robustness of DeR-Mamba under adversarial attack scenarios of varying intensities, specifically designed to simulate structural incompleteness and real-world noise, these scenarios mainly focus on altering graph structures and node features. Other types of adversarial attacks, such as those targeting edge attributes, introducing synthetic nodes, or tampering with interaction timestamps, have not yet been thoroughly explored. Furthermore, the proposed DeR-Mamba is currently tailored only for discrete-time dynamic graphs and lacks validation in the domain of continuous-time dynamic graphs. Regarding the interpretability of frequency decoupling, although visualization analysis demonstrates that AFDM effectively disentangles high- and low-frequency signals to filter and suppress noise propagation, the mapping between deep spectral features and specific graph structural semantics remains complex. It remains challenging to intuitively explain how the suppression or enhancement of specific frequency bands corresponds to specific types of adversarial attacks. In summary, future research will extend the applicability of DeR-Mamba, explore a wider range of adversarial attack scenarios, enhance interpretability in the spectral domain, and validate its effectiveness on continuous-time dynamic graphs.

