# OpenReview forum: "Entangled No More: Multi-Domain Decoupling for Robust Dynamic Graph Neural Networks"
_ICML.cc/2026/Conference — ICML 2026 regular_

### Official Review · Reviewer_LBnP · 2026-03-09

**Soundness:** 3
**Presentation:** 3
**Significance:** 3
**Originality:** 4
**Overall Recommendation:** 4
**Confidence:** 4

**Summary:**

This paper proposes DeR-Mamba, a framework for robust dynamic graph learning that decouples two entangled failure modes: representation drift from structural incompleteness and signal distortion from noise perturbations. The framework consists of MP-K²alman for spatial decoupling, AFDM for frequency-domain spectral purification, and a selective state-space mechanism to mitigate the temporal redundancy that accumulates from the two primary problems.

The problem motivation is solid, the experimental evaluation is thorough, and the theoretical results in Appendix B are non-trivial. These are genuine strengths. The primary concern, however, is that Section 4 consistently describes what each component does without explaining why it was designed that way. For a paper that introduces several non-standard technical choices (multi-particle kernel subspace sampling, Kalman-style confidence fusion, Softmax PRF mapping), the absence of design rationale makes the methodology difficult to evaluate and reproduce. The notational ambiguity around $R_q^{(m,h)}$ and $R_k^{(m,h)}$ compounds this problem. These are presentation issues rather than fundamental flaws, and addressing them does not require new experiments or theory. I lean toward acceptance, contingent on a revision that adds explicit motivation for the core design choices in Section 4.1 and resolves the notational inconsistency in W2.

**Compliance With Llm Reviewing Policy:**

Affirmed.

**Key Questions For Authors:**

See weeknesses.

**Limitations:**

yes

**Strengths And Weaknesses:**

## Strengths

- The multi-domain entanglement perspective is well-motivated. The argument that structural defects amplify spectral noise which then propagates temporally is clearly laid out and not commonly addressed in the dynamic graph robustness literature.
- The theoretical analysis is a genuine contribution. Theorem 1 establishes stability and optimality of the fused multi-particle representation via an information-theoretic argument, and Propositions 1-2 provide spectral non-expansion and perturbation stability guarantees for AFDM. These results support the empirical claims.
- The experimental evaluation is comprehensive: three diverse datasets spanning different temporal scales (16 years, 24 months, 30 days), multiple attack types, and a wide baseline pool.

## Weaknesses

**W1. The methodology section prioritizes describing how the components are designed over explaining why they are designed that way.** This issue runs throughout Section 4 but is most pronounced in Section 4.1. The paper presents MP-K²alman as sampling latent evolutionary paths in multiple kernel subspaces and fusing confidence across diverse particle views via Kalman-style updates, but neither choice is justified. Why are multiple kernel subspaces the appropriate representation for latent structural uncertainty under incomplete observations? Why is a Kalman-style confidence fusion, rather than, say, a simpler attention pooling, the right mechanism for combining particle views? The paper does not address these questions. Similarly, the decomposition into M particles and H attention heads is introduced as a parameterization without explaining what role each part serves: do the particles represent distinct structural hypotheses, and if so, how does this differ from what the attention heads capture? The role of the Softmax kernel approximation via Positive Random Features is also unclear. Eq. (1) defines a PRF mapping grounded in Mercer's theorem, but the paper does not explain what property of the Softmax kernel makes it suitable for this setting or what would be lost by using a different kernel. Section 4.2 and 4.3 are less severe but share the same pattern. Overall, the paper reads as a sequence of technical choices whose justification is either absent or delegated to appendices in a way that makes the main contribution difficult to evaluate.

**W2. The relationship between $R_q^{(m,h)}$, $R_k^{(m,h)}$ and the subsequent equations is unclear.** Section 4.1 introduces two random projection matrices $R_q^{(m,h)}$ and $R_k^{(m,h)} \in \mathbb{R}^{d \times d}$, but it is not specified where either matrix participates in computation. Eq. (1) uses a single matrix $R^{(m,h)}$ for the PRF mapping, and Eqs. (3)-(6) reference $\varphi_i = R^{(i,h)}$ without distinguishing a $q$-variant from a $k$-variant. The relationship between $R_q^{(m,h)}$, $R_k^{(m,h)}$ and this $R^{(m,h)}$ is never stated. Furthermore, whether $R_q^{(m,h)}$ and $R_k^{(m,h)}$ correspond to the Query and Value branches in Eq. (2) is also left implicit. The authors should explicitly state how these matrices are used in the computation and clarify their connection to the Q and V terms.

---

> ### Author Rebuttal · Authors · 2026-03-31
>
> We are profoundly grateful for your inspiring and constructive feedback. Your insightful suggestions have been genuinely enlightening for refining our presentation.
> > **Response W1**：
>
> > **W1-Q1: Why use multiple kernel subspaces for structural uncertainty under incomplete observations?**：
>
> - **The unreliability of single topological observations:** In dynamic graphs afflicted by inherent structural flaws or adversarial attacks, relying on a single graph topological observation is highly unreliable and inevitably incurs extremely high variance.
>
> - **Providing complementary structural hypotheses:** By mapping incomplete graphs into multiple continuous random feature subspaces, the model explores diverse topological trajectories in parallel. This provides robust distributional support for subsequent Kalman filtering, preventing representation drift stemming from a single corrupted view.
>
> > **W1-Q2: Why use Kalman-style updates to fuse particle views instead of simpler attention pooling?**：
>
> - **Vulnerability of Attention Pooling:** Standard attention relies on dot-product "mean" aggregation. Under attack, it is easily dominated by high-frequency anomalous edges, erroneously amplifying adversarial noise.
>
> - **Variance-Aware Kalman Fusion:** Grounded in the LMMSE principle, Kalman fusion evaluates confidence rigorously through variance and covariance. High-frequency perturbations statistically manifest as high variance. Consequently, the Kalman gain automatically assigns these corrupted views a near-zero weight, forming an uncertainty-aware isolation mechanism inherently unachievable by traditional attention.
>
> > **W1-Q3: In the $M$-particle and $H$-head parameterization, do particles still encode distinct structural hypotheses? How do they differ from attention heads?**
>
> They are responsible for capturing entirely distinct distributional characteristics during the decoupling process, and the two are orthogonal and complementary.
> - **Particles $M$:** Responsible for processing topological uncertainty. Each particle represents a potential trajectory of graph structural evolution, serving to defend against inherent structural flaws and adversarial attacks.
>
> - **Attention heads $H$:** Responsible for capturing semantic diversity. Similar to traditional Transformers, they map node features into different semantic subspaces.
>
> In essence: $M$ resolves the question of "whether an edge exists and is reliable," whereas $H$ resolves the question of "what specific semantic meaning the edge conveys."
>
> > **W1-Q4: What is the purpose of the Softmax kernel approximation via Positive Random Features?**
>
> - **Necessity of the Softmax kernel:** It provides crucial exponential decay, amplifying highly similar node pairs while rapidly decaying the weights of spurious connections to near zero.
>
> - **Detriment of alternative kernels:** Linear or polynomial kernels lack this capability, inevitably causing severe over-smoothing and failing to isolate spurious structural noise.
>
> - **Efficiency via PRF:** Exact full-graph Softmax computation requires an intractable $O(N^2)$ complexity. We utilize Positive Random Features (PRF) to approximate this non-linear kernel in $O(N)$ linear time, perfectly preserving its spatial filtering characteristics while satisfying the extreme efficiency demands of dynamic graphs.
>
> > **Response W2:**
>
> You have acutely pointed out the ambiguity caused by a notational oversight in Section 4.1. We fully accept your correction, as this is indeed a typographical error. Here, we explicitly clarify the precise mapping relationships among these matrices.
>
> > **W2-Q1: Clarification regarding $R_k^{(m,h)}$.**
>
> Unlike standard attention-based PRF approximations that require both $R_q$ and $R_k$, our MP-K²alman architecture does not contain an explicit Key (K) branch. As formulated in Eq. (2), we treat the Query as the Observation field and the Value as the latent State. The Value branch only undergoes a linear transformation without participating in the PRF mapping. Therefore, the presence of $R_k^{(m,h)}$ in the text is merely a clerical error carried over from traditional PRF notation and is not involved in any actual computations.
>
> > **W2-Q2: Relationship among the Matrices.**
>
> To eliminate any referential ambiguity, we clarify the specific correspondences:
> - **Eq. (1):** $R^{(m,h)}$ serves as the random feature projection matrix (drawn from $\mathcal{N}(0,\sigma^{-2}I)$ and orthogonalized) used to approximate the kernel function. In this general equation, it acts as a generic placeholder to introduce the PRF mechanism.
> - **Eq. (2):** The subscript $q$ in $\phi_q^{(m)}$ explicitly invokes the Query-specific matrix $R_q^{(m,h)}$. The Value ($\mathrm{V}$) branch involves no random feature matrix.
> - **Eqs. (3)-(6):** Since exclusively the Query is projected, the shorthand parameter set $\Pi_i = R^{(i,h)}$ strictly and uniquely refers to $R_q^{(i,h)}$.
>
> We have revised it in Line 163 and explained it in detail in Eq. (2).

---

> > ### Author Rebuttal · Reviewer_LBnP · 2026-04-04
> >
> > I thank the authors for their detailed response. After reading the rebuttal and the other reviewers' comments, I believe that my original assessment is consistent with the current quality of the paper.

---

> > > ### Author Response · Authors · 2026-04-07
> > >
> > > Thank you very much for your thoughtful review and for acknowledging our rebuttal. We are delighted that our responses regarding the methodological design rationale and notational clarifications effectively addressed your concerns. Your feedback has been invaluable, and we will carefully integrate all these detailed explanations into the revised manuscript.

---

### Official Review · Reviewer_Rnc9 · 2026-03-10

**Soundness:** 3
**Presentation:** 4
**Significance:** 3
**Originality:** 4
**Overall Recommendation:** 5
**Confidence:** 4

**Summary:**

This paper introduces DeR-Mamba(Decoupling for Robust Mamba), a framework designed to address representation drift and signal distortion in dynamic graphs. By decoupling spatiotemporal entanglement, the model integrates a Multi-Particle Kernel Kalman (MP-K$^2$alman) filter for structural estimation and an Adversarial-aware Frequency Decoupling Module (AFDM) for spectral purification. Leveraging the selective scanning of State Space Models (SSMs), DeR-Mamba demonstrates superior adversarial robustness over existing DGNNs.

**Compliance With Llm Reviewing Policy:**

Affirmed.

**Key Questions For Authors:**

+ Boundary Effects in AFDM: Equations (11-12) employ a centralized sliding window ($k_0 = \lfloor k/2 \rfloor$). However, at the final snapshot ($t=T$), future data is unavailable. Please clarify the padding strategy used in the implementation. Does this approach introduce edge noise or risk look-ahead bias that could compromise the prediction accuracy at $T+1$?
+ Motivation for Decoupling Order: The spatial decoupling (MP-$K^2$alman) is executed prior to spectral denoising (AFDM). Could this sequence allow high-frequency structural noise to propagate to neighboring nodes before it is purified? The authors should justify the theoretical motivation behind this specific execution order.
+ Notational Issues in Pseudocode: In Algorithm 1 (Line 17), the inter-graph structural matrix $\hat{\mathbf{A}}_{\text{inter}}^{1:T}$ is invoked without prior initialization in Lines 1–16. Please ensure all variables are properly defined and initialized.

**Limitations:**

It is noted that the authors provide a thorough discussion regarding the limitations of the framework in Appendix K.

**Strengths And Weaknesses:**

Strengths
+ Unlike existing methods that focus solely on spatial or temporal redundancy, this work incorporates wavelet-based spectral analysis, providing a more comprehensive dimension for noise resilience.
+ The integration of particle filtering with kernel approximation provides a rigorous theoretical foundation for characterizing uncertainty in dynamic graph learning.
+ While maintaining linear O(N) complexity, the framework achieves superior performance in Link Prediction tasks across multiple real-world datasets.

Weaknesses
+ In practice, structural incompleteness manifests directly as high-frequency distortion in the Graph Laplacian spectrum. Given this, why is it necessary to employ two distinct and computationally expensive mathematical toolsets instead of addressing these issues within a unified Graph Signal Processing framework?
+ The details regarding boundary handling at the sequence endpoints in the AFDM module are insufficient. This lack of clarity may compromise the mathematical rigor and predictive accuracy of the frequency-domain decoupling.
+ The design choice regarding the execution order of spatial and spectral decoupling lacks sufficient motivation. There is a concern that this specific sequence could lead to premature noise propagation.
+ The pseudocode contains minor notational inconsistencies, specifically missing variable definitions.

---

> ### Author Rebuttal · Authors · 2026-03-31
>
> We sincerely appreciate your recognition of our work and your constructive suggestions. We would be grateful if you could let us know whether your concerns have been addressed in our response.
>
> > **Response Q1: Boundary Effects in AFDM.**
>
> We would like to clarify that we employ a **Replication Padding strategy**, which fundamentally prevents the occurrence of the two issues you mentioned:
>
> - **Avoiding Look-ahead Bias:** We pad the sequence by replicating the features of the last visible snapshot, $\mathrm{Z}_T$, for $\lfloor (k-1)/2 \rfloor$ subsequent steps. This padding relies exclusively on historical data ($\le T$). It completely isolates any dependence on the actual distribution at time $T+1$ in both mathematical formulation and code logic, thereby strictly preserving causality without any look-ahead bias.
>
> - **Mitigating Edge Noise:** By utilizing replication padding, we implicitly assume that the system maintains its current stationary state for an extremely short window into the unknown future. This strategy ensures zero-order continuity of the signal at the boundary and drastically suppresses the high-frequency energy surges induced by boundary truncation. In contrast, if zero-padding were applied, the signal would suffer a drastic step-down at the boundary. The high-pass filter of AFDM would misinterpret this artificial drop as a severe adversarial structural mutation, resulting in significant edge noise. Consequently, our approach maximally guarantees the accuracy of AFDM's spectral decoupling at time $T$ while strictly adhering to causality.
>
> To provide readers with a clearer understanding of the underlying principles and mechanisms of AFDM, we have now incorporated this padding strategy for boundary effects into Section 4.2 of the Methodology, immediately following Eq. (12).
>
> > **Response Q2: Motivation for Decoupling Order.**
>
> The concern you raised, that premature spatial decoupling might cause high-frequency noise components to propagate early to adjacent nodes, is indeed a fatal flaw in some traditional graph convolutional models. On the contrary, in our framework, the sequence of performing spatial decoupling (MP-K²alman) prior to spectral decoupling (AFDM) not only prevents noise diffusion but is actually a Strict Causal Necessity dictated by underlying graph signal processing theory. The primary reasons are as follows:
>
> - **Blocking rather than diffusing: The Bayesian filtering mechanism of MP-K²alman.** Your concern primarily stems from the blind aggregation typical of traditional message passing. However, MP-K²alman is not a conventional spatial neighborhood aggregation, but rather a Bayesian observation field based on multi-particle sampling. When confronting structural noise, MP-K²alman generates the Kalman Gain as the ratio of Query-Value covariance to variance. Equivalent to confidence weighting, this assigns an extremely low Kalman Gain to anomalous, spurious structures with exceptionally high variance. Therefore, it acts as an isolation barrier at the first instance of spatial decoupling, proactively blocking high-frequency structural noise propagation to neighboring nodes. This lays a clean foundation for subsequent signal purification.
>
> - **Causal Dependency between Topology and Spectrum:** In graph signal processing theory, the definition of the spectral domain is inherently dependent on the underlying graph topology. In dynamic graphs suffering from structural missingness or adversarial attacks, the original graph Laplacian matrix itself is already severely corrupted. If AFDM spectral decomposition were to be performed first on this corrupted topology, the separated "low-frequency" and "high-frequency" components would suffer from semantic misalignment (i.e., low frequencies would no longer correspond to structural consistency, and high frequencies could not accurately represent noise perturbations). Spectral purification built upon such an erroneous basis would be rendered meaningless.
>
> We are deeply grateful for this question, which strikes at the core of our logic. We have added this theoretical motivation on the causality of the decoupling sequence to the beginning of Section 4.2.
>
> > **Response Q3: Notational Issues in Pseudocode**
>
> We would like to clarify that the inter-graph structural matrices $\hat{\mathrm{A}}_{\text{inter}}^{1:T}$ are indeed already listed in the "Require" section of Algorithm 2. However, we acknowledge an oversight in the top-level description of Algorithm 1, where we inadvertently omitted its explicit declaration as a preprocessing output.
>
> To address this, we have now added a clarification in the initialization section of Algorithm 1, explicitly stating that this matrix is computed based on the original sequence $\mathcal{G}^{1:T}$ during the preprocessing stage, prior to entering the training loop. Furthermore, we have conducted a thorough check to ensure that all variables are properly defined and initialized throughout the algorithms.

---

> > ### Author Rebuttal · Reviewer_Rnc9 · 2026-04-01
> >
> > Thanks for the rebuttal. The padding strategy (Q1) and the logic for the execution order (Q2) are both convincing. Overall, this is a solid work, and I have increased my score.

---

> > > ### Author Response · Authors · 2026-04-07
> > >
> > > Thank you very much for your positive feedback. We are delighted to hear that our explanations regarding the padding strategy and the logic for the execution order effectively addressed your questions.

---

### Official Review · Reviewer_ySnN · 2026-03-24

**Soundness:** 2
**Presentation:** 3
**Significance:** 2
**Originality:** 3
**Overall Recommendation:** 4
**Confidence:** 3

**Summary:**

This paper aims to address the lack of robustness in dynamic graph neural networks under multi-domain scenarios. The authors focus on two key challenges: structural incompleteness and noise perturbation. To this end, they propose DeR-Mamba, which incorporates a Multi-Particle Kernel Kalman observation field to model spatial structure and an adversarial-aware frequency decoupling mechanism to purify spectral signals. The authors conduct extensive experiments to validate the effectiveness of the proposed method.

**Compliance With Llm Reviewing Policy:**

Affirmed.

**Final Justification:**

The responses have fully addressed my concerns. I believe this is a good work with clear motivation and novel method. I'd like to raise my score to "Weak Accept (4)".

**Key Questions For Authors:**

Please see the above weaknesses.

**Limitations:**

yes.

**Strengths And Weaknesses:**

Strengths:

1. The motivation is clear. Figure 1 illustrates how structural incompleteness and noise perturbations propagate and accumulate over time, providing a clear rationale for the proposed decoupling strategy.

2. The theoretical analysis is well established.

3. The experimental evaluation is detailed and comprehensive.

Weaknesses:

1. Section 4.3 describes the SSM component, but it is unclear how much of the performance gain is attributable to the "selective state-space" versus the robust preprocessing by MP-K²alman and AFDM.

2. The authors place most of the theoretical analysis in the appendix. It would improve the paper if some core theorems and their corresponding intuitions were included in the main text.

3. It is unclear why the other baseline methods use two-layer GNNs, whereas DeR-Mamba is built upon a one-layer GNN. The authors should clarify whether this architectural difference affects the fairness of the comparison and the reported performance gains.

4. While the paper presents a well-motivated combination of structural uncertainty estimation, spectral decoupling, and state-space temporal modeling, each component is largely built upon existing ideas rather than introducing a fundamentally new modeling principle.

---

> ### Author Rebuttal · Authors · 2026-03-30
>
> We sincerely appreciate your careful review and constructive feedback. Below we summarize our revisions and would appreciate your confirmation on whether they address your concerns.
> > **Response W1**：
>
> To isolate the contribution of the selective state space, MP-K²alman, and AFDM, we conduct additional ablation studies on the SSM component, including replacing it with LSTM and Transformer, as shown in Table 1.
>
> **Table 1: Ablation isolating the contribution of the SSM on ACT.**
> | Model |Original|Struct. Attack|Feat.(λ=0.5)|Feat.(λ=1.0)|Feat.(λ=1.5)|Evas.(n=1)|Evas.(n=2)|Evas.(n=3)|Pois.(n=1)|Pois.(n=2)|Pois.(n=3) |
> |:---|:---:|:---:|:---:|:---:|:---:|:---:|:---:|:---:|:---:|:---:|:---:|
> |**DeR-Mamba**|**97.17**|**96.6**|**80.78**|**75.83**|**73.48**|**88.75**|**88.1**|**79.3**|**88.46**|**87.56**|**79.02**|
> | DeR-Mamba (w/o&nbsp;SSM) |95.84|95.16|80.13|75.32|72.73|87.48|86.9|78.27|87.36|86.81|78.19|
> | DeR-Mamba (LSTM) |96.46|95.86|80.28|75.55|72.83|86.25|85.47|76.24|86.25|85.45|76.23|
> | DeR-Mamba (Transformer) |96.68|96.15|80.00|75.20|72.39|86.86|86.2|76.74|86.87|86.16|76.70|
>
> This further demonstrates that:
>
> - **The necessity of filtering redundant information across time steps:**
> Removing the temporal-redundancy-aware SSM leads to a significant performance drop across all scenarios. This corroborates that the cumulative diffusion of temporal perturbation residuals during temporal propagation compromises the model's ability to learn robust representations.
>
> - **The superiority of the dynamic-graph-redundancy-aware SSM:**
> Under attacks, naive models (LSTM/Transformer) indiscriminately propagate perturbations, performing even worse than the w/o SSM baseline. Conversely, DGRS adaptively modulates $\Delta$ via inter-graph structures, enabling temporal redundancy-aware filtering to suppress rather than amplify disturbances.
>
> We have revised the manuscript accordingly and added these details to Appendix H.1.
>
>
> > **Response W2**：
>
> We fully agree with your constructive suggestion to incorporate certain theorems and corollaries into the main text. As you rightly pointed out from a reader’s perspective, we have now integrated the following corollaries into Sections 4.1 and 4.2 of the Methodology section:
>
> - **Information-Theoretic Stability & Optimality:** Explains how multi-particle filtering maintains a lower bound on information capacity under extreme observations.
> - **Spectral Non-Expansiveness:** Clarifies how AFDM suppresses non-stationary high-frequency noise without disrupting system dynamics.
>
> > **Response W3**：
>
> **Fairness of Layer Configuration:** Our setup follows optimal settings and standard practices (e.g., DG-Mamba, AAAI 2025).
>
> - **Theoretical Justification:** Baselines need 2 layers for a 2-hop receptive field, whereas DeR-Mamba requires just 1 because our MP-K²alman and AFDM already pre-aggregate multi-hop priors.
>
> - **Empirical Proof:** To strictly validate fairness, Table 2 aligns DeR-Mamba with top baselines under both 1-layer and 2-layer settings. DeR-Mamba consistently achieves the best performance regardless of the layer count.
>
> We have added these details to Appendix F.6.
>
> **Table 2: Comparison under 1-layer and 2-layer settings on ACT.**
>
> |Model|Original|Struct. Attack|Feat. (γ=0.5)|Feat. (γ=1.0)|Feat. (γ=1.5)|
> |:---|:---:|:---:|:---:|:---:|:---:|
> | DGIB-Cat (1-layer) |92.27|83.60|68.89|64.84|60.09|
> | DG-Mamba (1-layer) |96.69|96.09|79.17|73.53|68.96|
> | **DeR-Mamba (1-layer)** |**97.17**|**96.60**|**80.78**|**75.83**|**73.48**|
> | DGIB-Cat (2-layer) |94.89|88.27|73.92|68.88|65.99|
> | DG-Mamba (2-layer) |96.67|96.14|79.36|73.76|70.21|
> | **DeR-Mamba (2-layer)** |**97.33**|**96.79**|**81.00**|**76.53**|**74.25**|
>
> > **Response W4**：
>
> We completely understand your concerns and appreciate your validation of our motivation. However, DeR-Mamba is not merely a heuristic combination of existing ideas, but rather introduces a novel system-level paradigm to address "multi-domain entanglement," a core physical phenomenon in dynamic systems largely overlooked by prior work.
>
> - **Systematic decoupling against "error cascading":** Standing as the first framework to systematically analyze spatial, spectral, and temporal coupling for dynamic graph robustness , our paradigm moves beyond existing single-perspective methods to jointly resolve the complex interplay of structural flaws, spectral pollution, and temporal redundancy.
>
> - **MP-K²alman is not a traditional Kalman filter:** Marking the first application of Kalman filtering for dynamic graph robustness, it integrates Positive Random Features (PRF) to bypass the $O(N^2)$ attention bottleneck, achieving an $O(N)$ kernelized update for highly efficient structural uncertainty estimation.
>
> - **Beyond a naive vanilla Mamba:** A novel DGRS mechanism enables temporal redundancy-aware state evolution via adaptive modulation of $\Delta$ using $\hat{A}_{\text{inter}}^{1:T}$, effectively suppressing perturbation accumulation.

---

> > ### Author Rebuttal · Reviewer_ySnN · 2026-04-02
> >
> > Thanks for the responses and additional experiments. I have no further questions. I will raise my score.

---

> > > ### Author Response · Authors · 2026-04-07
> > >
> > > We sincerely thank you for your positive feedback and for taking the time to review our responses. We are glad that our responses have addressed your concerns, and we truly appreciate your recognition of our work.

---

### Decision · Program_Chairs · 2026-04-30

**Decision:**

Accept (regular)

**Comment:**

This paper proposes DeR-Mamba to address representation drift and signal distortion in dynamic graphs caused by tightly entangled spatial-temporal evolution. In the rebuttal, the authors provided additional ablation experiments and further clarification of the method, including the boundary effects of ADFM and the motivation for decoupling order. The experiments are sufficient, and the empirical results are promising.

The concersn of all reviewers are fully resolved. Three reviewers give positive recommendations (1 accept and 2 weak accept). Therefore, I recommend accepting this paper.

In the revision, the authors should update the additional ablation results and address the questions that were resolved during the rebuttal.